# E-mem: Multi-Agent Based Episodic Context Reconstruction for LLM Agent Memory

Kaixiang Wang [* ‡ 1]  Yidan Lin [* 1]  Zihan Wang [1]  Bunyod Suvonov [1]  Zhaojiacheng Zhou [1]  Yuxiang Zheng [1]
Jiaxi Cao [1]  Zhiheng Dong [1]  Chentao Wu [1 2 3]  Jiong Lou [† 1 2 3]  Jie LI [† 1 2 3]

## Abstract

The evolution of Large Language Model (LLM) agents towards System 2 reasoning, characterized by deliberative, high-precision problem-solving, requires maintaining rigorous logical integrity over extended horizons. However, prevalent memory preprocessing paradigms suffer from destructive de-contextualization. By compressing complex sequential dependencies into pre-defined structures (e.g., embeddings or graphs), these methods sever the contextual integrity essential for deep reasoning. To address this, we propose E-mem, a framework shifting from memory preprocessing to episodic context reconstruction inspired by biological engrams. E-mem employs a heterogeneous hierarchical architecture where multiple assistant agents maintain uncompressed memory contexts, while a central master agent orchestrates global planning. Unlike passive retrieval, our mechanism empowers assistants to locally reason within activated segments, extracting context-aware evidence before aggregation. Evaluations on the LoCoMo benchmark demonstrate that E-mem achieves over 54% F1, surpassing the state-of-the-art GAM by 7.75%, while reducing token cost by over 70%. Our work is available on https://github.com/dog-last/E-mem.

## 1. Introduction

Large Language Models (LLMs) have evolved from stochastic text generators into the central cognitive controllers of

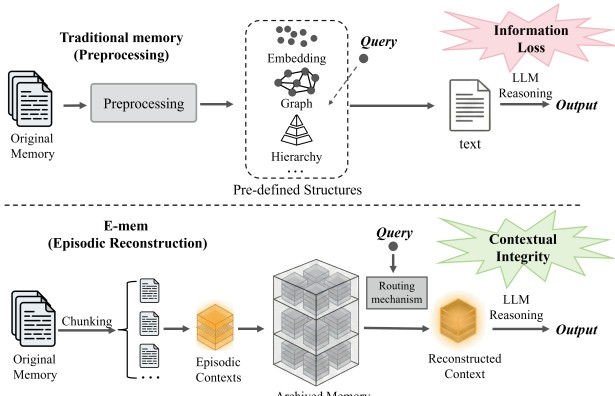

*Figure 1.* Traditional Memory System vs. E-mem

autonomous agents (Xi et al., 2025; Wang et al., 2024; Mei et al., 2025). Empowered by advanced planning capabilities and external tool integration (Mei et al., 2025), these systems are now transitioning towards System 2 reasoning—characterized by deliberative, sequential problem-solving in dynamic environments (Zhang et al., 2022; Yao et al., 2022; Schick et al., 2023). However, supporting this shift demands rigorous adherence to causal chains. In such scenarios, maintaining extensive history (Zheng et al., 2026) becomes pivotal to preserving the logical integrity essential for deep, long-horizon planning (Park et al., 2023; Wang et al., 2023; Shinn et al., 2023).

However, expanding operational horizons presents significant challenges. Merely extending context windows often triggers the "Lost-in-the-Middle" phenomenon (Liu et al., 2024), explicitly necessitating robust memory management mechanisms (Zhang et al., 2025b; Packer et al., 2023). Existing approaches primarily rely on preprocessing to index memory, mapping raw, unstructured contexts into pre-defined structures (e.g., static embeddings, knowledge graphs, or hierarchical archives) (Xu et al., 2025; Packer et al., 2023; Hu et al., 2023). While enabling efficient lookup, this strategy results in destructive de-contextualization: by compressing complex sequential dependencies into rigid representations, it disrupts the critical integrity required for deep reasoning (as shown at the top of Figure 1) (Sarthi et al., 2024; Jimenez Gutierrez et al.,

[1] School of Computer Science, Shanghai Jiao Tong University, Shanghai, China. [2] State Key Laboratory of Digital Finance (In Preparation). [3] SJTU Suzhou Innovation Institute. *: Co-first authors. ‡: Primary contact to Kaixiang Wang: wangkaxiang@sjtu.edu.cn. †: Correspondence authors. Correspondence to: Jiong Lou <lj1994@sjtu.edu.cn>, Jie Li <lijiecs@sjtu.edu.cn>.

*Proceedings of the 43rd International Conference on Machine Learning*, Seoul, South Korea. PMLR 306, 2026. Copyright 2026 by the author(s).

2024). Consequently, such methods struggle to reconstruct complex causal chains or comprehend memories within their original sequential contexts, ultimately yielding suboptimal performance on information-dense benchmarks like LoCoMo (Maharana et al., 2024).

To address these limitations and strictly ensure logical deduction over memory, we introduce E-mem. This framework transitions from memory preprocessing to episodic context reconstruction (as shown at the bottom of Figure 1). **(i)** Inspired by biological engrams, E-mem preserves the full episodic context of original experiences, enabling the active re-experiencing of past events. **(ii)** To mitigate the information distortion inherent in ultra-long contexts while ensuring cost-effectiveness for scalable deployment, E-mem adopts a heterogeneous hierarchical master-assistant architecture. In this design, a central master agent orchestrates global planning, while multiple assistant agents (implemented as small language models, SLMs) serve as memory units. Each assistant agent maintains the raw memory context of a specific segment. **(iii)** For each query, a routing mechanism selectively activates a relevant subset of assistant agents. Crucially, instead of merely retrieving text chunks, these agents execute episodic context reconstruction—a process where they actively re-experience and reason within the restored native contexts to derive precise, local evidence for the master agent. This mechanism not only mitigates context window constraints and reduces computational costs but also ensures high-fidelity, context-preserving inference, offering a distinct advantage in complex, multi-hop reasoning tasks where traditional retrieval mechanisms fall short.

Empirical evaluations on LoCoMo (Maharana et al., 2024) and HotpotQA (Yang et al., 2018) confirm that E-mem achieves state-of-the-art performance. The system delivers superior accuracy compared to SOTA baselines while significantly reducing token cost, validating the efficiency of the proposed episodic context reconstruction paradigm.

In summary, our main contributions are as follows:

- **Episodic Context Reconstruction.** To address the destructive de-contextualization inherent in traditional memory preprocessing, we introduce E-mem, a framework that centered on episodic context reconstruction. Unlike static retrieval methods that sever sequential dependencies, our approach delegates active reasoning to assistant agents. They preserve and process the full context of memory segments locally, ensuring that only logically deduced evidence—rather than raw, noisy fragments—is surfaced to master agents.

- **Heterogeneous Hierarchical Master-Assistant Architecture.** We propose E-mem, a scalable framework that decouples high-level planning from memory retention. By coordinating a master agent with lightweight,

SLM-based assistant agents, our design mitigates the information loss inherent in preprocessing, ensuring high-fidelity reasoning across long horizons without suffering from the "lost-in-the-middle" phenomenon.

- **SOTA Performance with Token Efficiency.** Extensive evaluations on LoCoMo and HotpotQA demonstrate that E-mem outperforms strong baselines by an average of 7.75% (F1), with notable gains in complex multi-hop (+8.56%) and temporal reasoning (+8.87%). Crucially, these improvements are realized while reducing token cost by over 70%. These results confirm E-mem as a vital complement to traditional memory paradigms for System 2 reasoning.

## 2. Related Work

**Retrieval-Augmented Generation.** RAG has established itself as a fundamental paradigm to mitigate LLM hallucination and knowledge obsolescence by grounding generation in external corpora (Lewis et al., 2020; Karpukhin et al., 2020; Guu et al., 2020). While standard frameworks typically employ a vector-based "retrieve-then-generate" pipeline, recent *Agentic RAG* grants systems greater autonomy in retrieval planning and iterative context refinement (Jiang et al., 2023; Asai et al., 2024; Trivedi et al., 2023; Yao et al., 2022). Despite these advancements in retrieval logic, the underlying storage mechanism remains predominantly reliant on preprocessing. This approach inherently compresses rich sequential contexts into fixed geometric points, risking the loss of fine-grained sequential dependencies essential for reasoning (Liu et al., 2024; Wu et al., 2022; Bulatov et al., 2022).

**Memory Systems for Autonomous Agents.** Beyond simple retrieval, persistent and adaptive memory systems are critical for agents to maintain long-term interaction coherence. Addressing the finite context window, MemGPT (Packer et al., 2023) employs an operating system-inspired virtual context management technique via hierarchical paging. However, this paradigm relies on swapping fragmented chunks, necessitating redundant re-processing to restore sequential dependencies. Other works focus on optimizing memory structure and active evolution. For instance, G-Memory (Zhang et al., 2025a) introduces a graph-based hierarchical structure to enable navigation between global macro-views and local micro-interactions. Similarly, A-Mem (Xu et al., 2025) adopts a self-evolving framework based on the Zettelkasten method, while GAM (Yan et al., 2025) and ReasoningBank leverage multi-agent deep research and reasoning trajectory storage, respectively. Although recent efforts extend to personalization (Mem0 (Chhikara et al., 2025)) and benchmarking (MemoryBench (Ai et al., 2025)), these approaches remain bound to text-based preprocessing paradigms. By compressing

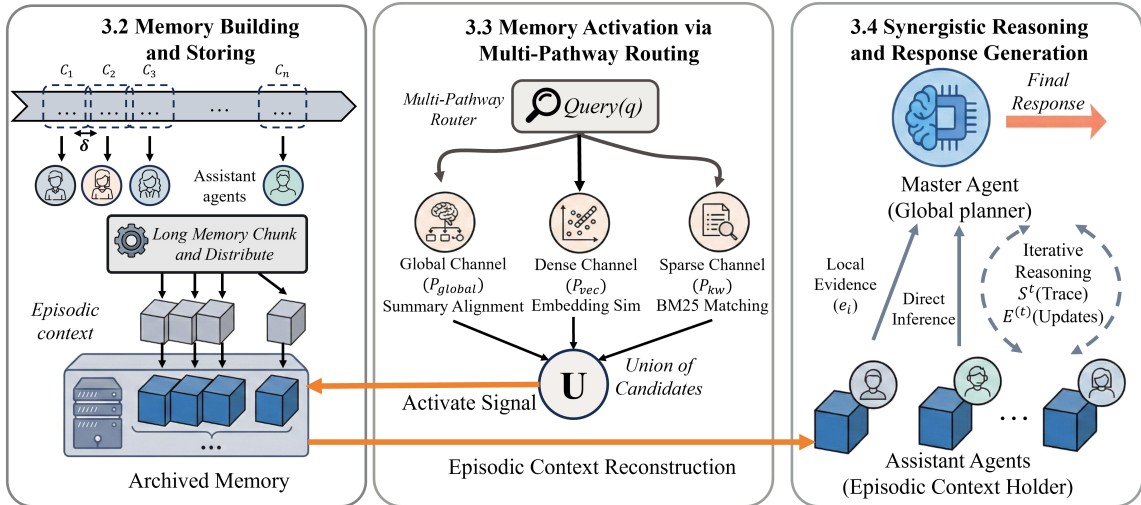

*Figure 2.* Overview of E-mem

complex contexts into rigid structures, they often disrupt the contextual integrity required for deep reasoning.

In contrast, E-mem introduces episodic context reconstruction, which ensures seamless inference integrity and serves as a critical complement to existing paradigms for high-precision, complex reasoning tasks.

## 3. Method

Cognitive science defines memory as the re-experience of intact episodic contexts rather than static retrieval (Tulving, 2002). In contrast, prevalent preprocessing paradigms force dynamic inputs into fixed structures, resulting in destructive de-contextualization. This rigid compression disrupts sequential dependencies, severing the contextual integrity essential for deep reasoning.

We propose E-mem, a framework centered on episodic context reconstruction, designed to explicitly preserve uncompressed memory segments and their inherent sequential dependencies (as shown in Figure 2). Implemented via a heterogeneous hierarchical architecture, the system functions through a streamlined three-stage process: first, a routing mechanism performs coarse-grained localization to selectively activate relevant archived memory units; subsequently, multiple assistant agents execute parallel fine-grained reasoning within these raw contexts to derive specific evidence; finally, the central master agent aggregates these distributed insights into a coherent, synergistic response.

### 3.1. Architecture

We propose E-mem, a heterogeneous hierarchical architecture designed to scale long-context reasoning by decoupling high-level planning from low-level memory retention. For-

mally, the system is defined as a collaborative tuple:

$$\mathcal{F} = \langle \mathcal{A}_{\text{master}}, \{\mathcal{A}_{\text{asst}}^{(i)}\}_{i=1}^{N}, \mathcal{R} \rangle, \tag{1}$$

where $\mathcal{A}_{\text{master}}$ acts as the central planner, $\{\mathcal{A}_{\text{asst}}^{(i)}\}$ represents a dynamic set of multiple assistant agents, and $\mathcal{R}$ denotes the multi-pathway semantic routing mechanism.

**Master Agent ($\mathcal{A}_{\text{master}}$): Global Planner and Synthesizer**. The master agent functions as the central orchestrator, decoupled from the burden of raw memory retention. Its primary role is to execute high-level cognitive planning and synthesize distributed evidence into a coherent response. Rather than processing the extensive raw context directly—which would incur prohibitive computational costs—the master agent operates in a sparse global planning space. It delegates the specific memory activation and localization tasks to the routing mechanism and interacts exclusively with the local evidence from activated memory units. Formally, we define the master agent's operation as a mapping function:

$$R = \mathcal{A}_{\text{master}}\left(q, \{e_i \mid i \in \mathcal{A}^*\}\right), \tag{2}$$

where $\{e_i\}$ denotes the set of local evidence tuples derived by the selected assistants. This design ensures that the reasoning process remains computationally tractable and focused on logical deduction, even as the scale of the underlying memory archive expands indefinitely.

**Assistant Agents ($\mathcal{A}_{\text{asst}}^{(i)}$): Episodic Context Holders**. The assistant agents serve as the memory units for parallelized storage and execution. To ensure deployment feasibility and scalability, we instantiate these agents using SLMs. Uniquely, each assistant $\mathcal{A}_{\text{asst}}^{(i)}$ employs a dual-representation strategy: it preserves the immutable, complete episodic context $\mathcal{E}_i$ for fine-grained local reasoning, while also maintaining a concise semantic summary $s_i$ used for global routing.

We formalize this composite context $\mathcal{S}_i$ as a tuple:

$$\mathcal{S}_i = \langle \mathcal{E}_i, s_i \rangle, \tag{3}$$

where $\mathcal{E}_i$ represents the high-fidelity raw token sequence and $s_i$ denotes the lightweight summary. This architecture allows the system to store extensive histories hierarchically while selectively reconstructing only the most relevant contexts. Formally, only the subset of agents identified by the routing mechanism (denoted as $\mathcal{A}^*$) is transitioned to the active inference path. This mechanism ensures high-fidelity reasoning by enabling the selected agents to re-experience the original memory context without noise.

**Multi-Pathway Routing Mechanism** ($\mathcal{R}$). To efficiently filter the memory archive, the system employs a dedicated routing mechanism that goes beyond simple summarization. Given a query $q$, this mechanism executes a multi-pathway policy $\pi$ to generate an activation distribution:

$$\mathcal{P}_{act} = \pi(q \mid \mathbf{S}, \mathcal{R}) \in [0,1]^N, \tag{4}$$

where $\mathcal{R}$ synthesizes heterogeneous signals, including global narrative alignment derived from pre-computed summaries $\mathbf{S} = \{s_1, \ldots, s_N\}$, symbolic entity triggers, and latent semantic vector associations. This design enables scalable, retrieval-based memory access without incurring runtime LLM inference overhead.

### 3.2. Memory Building and Storing

To transform the unbounded input stream into manageable memory contexts, E-mem implements a block-wise handling strategy. This approach is designed to preserve the full episodic contexts, avoiding the information loss typical of preprocessing.

**Sliding Window Segmentation with Overlap.** Given an unbounded input stream $\mathcal{X} = (x_1, x_2, \ldots)$, we employ a sliding window strategy to partition it directly into a sequence of discrete episodic contexts $\mathbf{E} = \{\mathcal{E}_1, \mathcal{E}_2, \ldots, \mathcal{E}_N\}$. With a window length $L$ and stride $S < L$, we introduce an overlap $\delta = L - S$ to preserve local sequential dependencies across boundaries. The $i$-th context is formally defined as:

$$\mathcal{E}_i = \{x_t \mid (i-1)S < t \le (i-1)S + L\}. \tag{5}$$

This overlap buffer ensures that tokens at the segment edges retain their immediate predecessor context, thereby maintaining semantic coherence during the routing and reconstruction phases.

**Episodic Memory Context Retention and Isolation.** Each $\mathcal{E}_i$ explicitly encapsulates the original uncompressed tokens and is maintained by a dedicated assistant agent $\mathcal{A}_{\text{asst}}^{(i)}$ as a standalone memory unit. Functioning primarily as Archived Memory, these units remain in a dormant state by default.

They are selectively transitioned to an active state for immediate inference only when explicitly triggered by the routing mechanism. This retention of the full context enables the system to resume generation directly from original data, ensuring strict logical integrity during reasoning.

**Incremental Updates.** E-mem supports efficient $O(1)$ streaming updates. New incoming tokens $\Delta x$ are directly appended to the active agent's *current episodic context* $\mathcal{E}_{\text{active}}$ via standard auto-regression, extending the memory buffer: $\mathcal{E}' \leftarrow \mathcal{E}_{\text{active}} \cup \{\Delta x\}$. When the capacity $L$ is reached, the context is solidified as a completed memory unit. A new agent $\mathcal{A}_{N+1}$ is instantiated by carrying over the overlap region to maintain context flow:

$$\mathcal{E}_{N+1}^{\text{init}} = \text{Extract}(\mathcal{E}_N, \text{overlap} = \delta). \tag{6}$$

This context transfer acts as a continuity bridge, ensuring that sequential flow remains uninterrupted and logically consistent as the system scales linearly.

### 3.3. Memory Activation via Multi-Pathway Routing

E-mem formulates activation as a Hierarchical Associative Routing process, which performs coarse-grained localization to selectively transition archived memory units to active inference. To accommodate the multifaceted nature of recall—ranging from broad narrative intents to precise entity details—we introduce a Multi-Pathway Activation framework. This mechanism routes the input query $q$ through three orthogonal signaling pathways operating in parallel:

- **Global Alignment** ($\mathcal{P}_{\text{global}}$): Activates memories via *macroscopic narrative anchoring*. By efficiently computing the dense vector similarity and sparse lexical alignment between the query and the concise semantic summaries ($s_i$), this pathway leverages the "frozen reasoning" of the summarization phase. It functions as a high-pass semantic filter, capturing the user's broader intent to exclude irrelevant noise and identify logically relevant segments.

- **Semantic Association** ($\mathcal{P}_{\text{vec}}$): Activates memories based on implicit latent alignment between the query and the full episodic context ($\mathcal{E}_i$). Unlike the summary-based global path, this pathway utilizes high-dimensional vector similarity against the raw chunk embeddings. It serves as a robust failsafe to identify chunks that resonate with the query's abstract intent, specifically compensating for cases where critical semantic nuances may have been lost or distorted during the summarization process.

- **Symbolic Trigger** ($\mathcal{P}_{\text{kw}}$): Activates memories via explicit entity matching between the query and the *original raw text content*. Analogous to how a specific name

triggers a flashback, this pathway employs sparse retrieval (e.g., BM25) to detect exact lexical overlaps. This ensures the high-precision recall of unique factual anchors (such as specific IDs or names) that might be omitted in the high-level summaries, guaranteeing that fine-grained details are not overlooked.

**Episodic Context Reconstruction via Activation Union.**
To ensure comprehensive routing we employ a Multi-Source Activation Union strategy. A memory unit $\mathcal{A}_{\text{asst}}^{(i)}$ undergoes an episodic memory transition from archived to active (inference-ready) if it receives a valid activation signal from any pathway. Formally, the set of activated agents $\mathcal{A}^*$ is defined as:

$$\mathcal{A}^* = \{\mathcal{A}_{\text{asst}}^{(i)} \mid \mathcal{A}_{\text{asst}}^{(i)} \in \mathcal{P}_{\text{global}} \vee \mathcal{A}_{\text{asst}}^{(i)} \in \mathcal{P}_{\text{vec}} \vee \mathcal{A}_{\text{asst}}^{(i)} \in \mathcal{P}_{\text{kw}}\}. \tag{7}$$

Notably, the total number of activated memory chunks, denoted as $k$, can be flexibly adjusted according to requirements of the task. This mechanism ensures that the system aggregates all potentially relevant contexts re-integrating for the subsequent reasoning phase, strictly preserving the integrity of the information required for complex inference.

### 3.4. Synergistic Reasoning and Response Generation

Upon identifying the candidate set $\mathcal{A}^*$, the system transitions to the phase of parallel episodic context reconstruction. While the router performs coarse-grained localization, the master agent delegates the specific inference burden to the assistant agents for fine-grained local reasoning. In this process, each activated assistant agent $\mathcal{A}_{\text{asst}}^{(i)}$ does not merely retrieve data but actively generates a response by attending to its preserved episodic context $\mathcal{E}_i$. Crucially, to ensure causal consistency, we enforce Temporal Anchoring in the reasoning output. The assistant scrutinizes the original sequential dependencies to derive a temporally grounded evidence tuple $e_i$, modeled as:

$$e_i = \langle c_i, \tau_i \rangle = \Phi_{\text{asst}}(q \mid \mathcal{E}_i), \tag{8}$$

where $c_i$ represents the deduced semantic evidence, and $\tau_i$ denotes the absolute timestamp corresponding to the identified event or state transition. This mechanism ensures logical integrity by re-experiencing the raw memory, enabling the extraction of subtle evidence accompanied by its precise temporal occurrence. By explicitly associating facts with their time of occurrence, the system empowers the master agent to resolve conflicting states (e.g., object displacement) during the subsequent aggregation phase. We categorize this collaborative reasoning process into two modes:

**Direct Inference.** Tailored for queries necessitating explicit fact retrieval, this mode focuses on synthesizing the evidence set $\mathbf{E} = \{e_i\}_{i \in \mathcal{A}^*}$ generated by the assistants. The master agent acts as a central reasoner, aggregating these local evidence tuples to derive a final response $R$. This global synthesis is formally modeled as:

$$R = \Psi_{\text{master}}(q, \mathbf{E}), \tag{9}$$

where $\Psi_{\text{master}}$ denotes the high-level reasoning function. Unlike simple concatenation, the master agent actively resolves semantic conflicts by enforcing chronological logic upon the given evidence. Specifically, it compares the absolute timestamps $\tau$ associated with each evidence unit to reconcile state discrepancies (e.g., location changes), prioritizing the most recent information to construct a cohesive and accurate logical chain. This hierarchical decoupling ensures rigorous logical integrity while maximizing scalability through parallelized context processing.

**Iterative Reasoning.** Beyond single-pass retrieval, the E-mem architecture is naturally extensible to an iterative Refine-and-Query framework for complex tasks requiring sequential deduction. In this operational protocol, the master agent maintains a dynamic reasoning trace $S^{(t)}$. At each step, should the current information be insufficient, the master agent formulates a targeted sub-query:

$$q^{(t)} = \pi_{\text{plan}}(q_{\text{init}}, S^{(t-1)}), \tag{10}$$

to probe relevant assistant agents. These assistants execute local inference over their encapsulated episodic contexts $\mathcal{E}_i$, yielding context-aware evidence $e_i^{(t)} = \Phi_{\text{asst}}(q^{(t)} \mid \mathcal{E}_i)$. The master agent then aggregates these findings to update the global trace $S^{(t)} \leftarrow \text{Agg}(S^{(t-1)}, \{e_i^{(t)}\})$. This cycle facilitates principled multi-step reasoning, terminating upon either trace convergence or a predefined iteration limit.

## 4. Experiments

We evaluate E-mem against state-of-the-art memory mechanisms for LLM agents, focusing on performance, robustness, and cost. Our analysis addresses three core questions:

**RQ1: Comparative Efficacy.** How does E-mem compare against state-of-the-art RAG and agentic memory systems on complex long-context reasoning?

**RQ2: Component & Scaling Analysis.** How do specific architectural components and backbone model sizes impact the system's overall effectiveness?

**RQ3: Cost-Efficiency.** Does E-mem achieve a superior trade-off between token consumption and reasoning performance compared to baselines?

### 4.1. Experiment Settings

**Datasets**. We evaluate E-mem against competitive baselines on two benchmarks: **LoCoMo (Maharana et al., 2024)**: A

*Table 1.* Results on LoCoMo Benchmark, compared with other outstanding baselines

| | | Overall | | Single-Hop | | Multi-Hop | | Temporal | | Open Domain | |
|---|---|---|---|---|---|---|---|---|---|---|---|
| | | F1 | BLEU-1 | F1 | BLEU-1 | F1 | BLEU-1 | F1 | BLEU-1 | F1 | BLEU-1 |
| **GPT-4o-mini** | **Long-Context** | 37.31 | 29.57 | 46.68 | 37.54 | 29.23 | 22.76 | 25.97 | 19.42 | 16.87 | 13.70 |
| | **RAG** | 44.73 | 39.40 | 52.45 | 47.94 | 27.50 | 20.13 | 46.07 | 40.35 | 23.23 | 17.94 |
| | **A-mem** | 39.65 | 32.31 | 44.65 | 37.06 | 27.02 | 20.09 | 45.85 | 36.67 | 12.14 | 12.01 |
| | **Mem0** | 45.10 | 35.08 | 47.65 | 37.82 | 38.72 | 27.13 | 48.93 | 40.15 | **28.64** | **21.58** |
| | **MEMORYOS** | 42.84 | 35.54 | 48.62 | 42.99 | 35.27 | 25.22 | 41.15 | 30.76 | 20.02 | 16.52 |
| | **LIGHTMEM** | 38.44 | 34.37 | 41.79 | 37.83 | 29.78 | 24.90 | 43.71 | 39.72 | 16.89 | 13.92 |
| | **GAM** | 45.31 | 37.78 | 47.74 | 40.90 | 34.84 | 27.72 | 53.91 | 43.93 | 26.03 | 19.48 |
| | **E-mem** | **54.17** | **44.34** | **59.23** | **50.58** | **42.64** | **34.38** | **59.82** | **44.57** | 24.89 | 18.15 |
| **Qwen 2.5-14B** | **Long-Context** | 38.31 | 31.89 | 46.05 | 39.56 | 32.08 | 24.46 | 30.51 | 24.45 | 14.89 | 11.41 |
| | **RAG** | 38.27 | 33.07 | 47.87 | 42.89 | 26.38 | 19.54 | 30.78 | 25.97 | 14.16 | 10.52 |
| | **A-mem** | 28.98 | 24.47 | 33.75 | 30.04 | 22.09 | 15.28 | 27.19 | 22.05 | 13.49 | 10.74 |
| | **Mem0** | 36.04 | 29.91 | 42.58 | 35.15 | 31.73 | 24.82 | 28.96 | 26.24 | 15.03 | 11.28 |
| | **MEMORYOS** | 40.28 | 34.72 | 46.33 | 41.32 | 38.19 | 29.26 | 32.24 | 27.86 | 20.27 | 15.94 |
| | **LIGHTMEM** | 31.39 | 27.15 | 34.92 | 31.22 | 25.45 | 19.61 | 32.03 | 27.70 | 15.81 | 11.81 |
| | **GAM** | 50.41 | 43.48 | 56.35 | 51.07 | 38.94 | 28.55 | 53.76 | 50.01 | 20.84 | 15.09 |
| | **E-mem** | **57.04** | **46.75** | **61.14** | **52.50** | **49.15** | **34.87** | **63.59** | **50.61** | **22.38** | **18.41** |

*Table 2.* F1 Score on HotpotQA Benchmark

| | GPT4o-mini | | | Qwen2.5-14B | | |
|---|---|---|---|---|---|---|
| **HotpotQA** | 400 | 800 | 1600 | 400 | 800 | 1600 |
| | F1 | F1 | F1 | F1 | F1 | F1 |
| **Long-Context** | 56.56 | 49.71 | 53.92 | 49.75 | 46.82 | 43.17 |
| **RAG** | 52.71 | 51.84 | 54.01 | 51.81 | 46.72 | 48.36 |
| **A-mem** | 33.90 | 30.22 | 31.37 | 27.04 | 25.65 | 22.92 |
| **Mem0** | 32.85 | 31.74 | 27.41 | 30.12 | 32.44 | 26.55 |
| **MEMORYOS** | 26.47 | 23.10 | 24.16 | 24.58 | 30.25 | 23.13 |
| **LIGHTMEM** | 40.93 | 35.28 | 30.02 | 37.30 | 27.72 | 28.25 |
| **GAM** | 54.75 | 52.86 | 53.71 | 48.40 | 41.10 | 44.32 |
| **E-mem** | **61.46** | **55.46** | **55.76** | **61.13** | **47.91** | **54.87** |

*Table 3.* Hallucination analysis of E-mem on the LoCoMo adversarial subset.

| assistant agent | | | master agent | | |
|---|---|---|---|---|---|
| | F1 | BLEU-1 | | F1 | BLEU-1 |
| **Qwen3-0.6B** | 85.11 | 80.14 | **GPT4o-mini** | 89.94 | 85.51 |
| **Qwen3-1.7B** | 87.31 | 83.17 | **GPT4o** | 89.37 | 86.52 |
| **Qwen3-4B** | 89.94 | 85.51 | **Gemini2.5-flash** | 93.62 | 90.3 |
| **Qwen3-8B** | 95.74 | 88.09 | **DeepseekV3** | 75.87 | 73.4 |
| **Qwen3-14B** | 95.03 | 88.06 | **Grok4-fast** | 77.80 | 75.53 |

ory architectures (e.g., hierarchy or graph) to actively curate historical information, aiming to enhance precision and efficiency in downstream tasks.

**Implementation details.** Experiments were conducted on four NVIDIA RTX 4090 GPUs. We instantiate E-mem with GPT-4o-mini and Qwen2.5-14B as master agents, supported by a set of Qwen3-4B assistant agents. To ensure a fair comparison, E-mem and all baselines utilize the same master LLM backbone and are restricted to the same number of retrieval rounds. Evaluation is performed using the F1 score and BLEU-1 metrics.

### 4.2. Performance Analysis

**Results on LoCoMo Benchmark.** Table 1 details the LoCoMo evaluation, where E-mem consistently establishes a new state-of-the-art across diverse backbones. Specifically, it surpasses the strongest baseline (GAM) by substantial margins, achieving +8.86% on GPT-4o-mini and +6.63% on Qwen2.5-14B in overall F1 scores. This performance advantage is critical in complex reasoning tasks like Multi-hop and Temporal subsets, where standard RAG often falters due to severe context fragmentation. In contrast, E-mem significantly boosts Multi-hop F1 by over 10 points (49.15% vs. 38.94%) on the Qwen backbone. This confirms that our episodic context reconstruction paradigm effectively preserves the autoregressive dependencies typically lost in

benchmark assessing long-term memory in multi-session dialogues. It evaluates coherence across long horizons via five sub-tasks: single-hop retrieval, multi-hop reasoning, temporal understanding, open-ended generation, and adversarial tasks. **HotpotQA (Yang et al., 2018)**: A Wikipedia-based multi-hop QA dataset. To test scalability with ultra-long contexts (over 200K tokens), we adapt it into a streaming setting across three scales (400, 800, and 1600 documents), stress-testing evidence recall from extensive archives.

**Baselines.** We benchmark E-mem against two categories: **Memory-Free Baselines.** Direct context processing without explicit maintenance. (1) Long-Context Windowing: Uses a sliding window to segment history into chunks processed independently, selecting the highest-confidence output as the answer. (2) Standard RAG: Retrieves the top-$k$ ($k = 20$) relevant segments via dense vector similarity to augment generation. **Memory-Based Systems.** We compare against cutting-edge memory-augmented agent frameworks established in 2024–2025, including A-Mem (Xu et al., 2025), Mem0 (Chhikara et al., 2025), MemoryOS (Kang et al., 2025), LightMem (Fang et al., 2025), and GAM (Yan et al., 2025). These methods construct specialized external mem-

*Table 4.* Model ablation study comparing the performance of different master and assistant models on the LoCoMo conversation 1.

|  |  | Overall | | Single Hop | | Multi-Hop | | Temporal | | Open Domain | |
| --- | --- | --- | --- | --- | --- | --- | --- | --- | --- | --- | --- |
|  |  | F1 | BLEU-1 | F1 | BLEU-1 | F1 | BLEU-1 | F1 | BLEU-1 | F1 | BLEU-1 |
| **Assistant Agent Model** | **Qwen3-0.6B** | 28.89 | 22.26 | 26.40 | 21.19 | 17.32 | 18.35 | 46.44 | 33.49 | 20.77 | 12.75 |
|  | **Qwen3-1.7B** | 44.78 | 35.10 | 40.82 | 34.30 | 35.64 | 29.53 | 65.03 | 48.63 | 31.02 | 14.66 |
|  | **Qwen3-4B** | 50.70 | 40.93 | 49.46 | 41.21 | 42.66 | 35.38 | 66.91 | 51.45 | 31.08 | 23.15 |
|  | **Qwen3-8B** | 49.80 | 40.94 | 45.98 | 37.05 | 52.11 | 47.45 | 62.28 | 49.29 | 29.11 | 22.11 |
|  | **Qwen3-14B** | 50.02 | 40.27 | 49.96 | 41.43 | 46.38 | 37.25 | 64.53 | 51.66 | 30.32 | 24.41 |
| **Master Agent Model** | **GPT-4o-mini** | 50.70 | 40.93 | 49.46 | 41.21 | 42.66 | 35.38 | 66.91 | 51.45 | 31.08 | 23.15 |
|  | **GPT-4o** | 51.73 | 43.70 | 51.04 | 43.04 | 45.88 | 42.37 | 64.77 | 52.83 | 32.74 | 24.53 |
|  | **Gemini2.5-flash** | 49.05 | 41.15 | 49.57 | 41.26 | 38.20 | 34.17 | 65.87 | 54.57 | 25.08 | 19.53 |
|  | **Grok4-fast** | 46.34 | 36.13 | 46.75 | 36.87 | 35.65 | 26.72 | 65.34 | 52.29 | 30.28 | 20.51 |
|  | **Deepseekv3** | 50.88 | 42.25 | 51.80 | 43.27 | 43.16 | 37.13 | 62.64 | 51.07 | 31.42 | 24.30 |

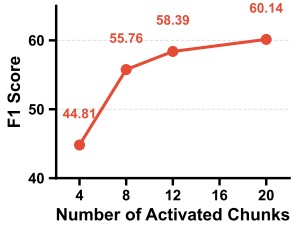

*(a)* Impact of the Number of Routed Chunks on Performance

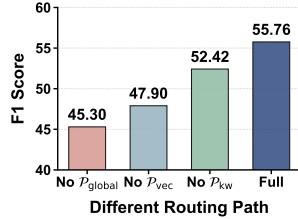

*(b)* Impact of Routing Pathways on System Performance

*Figure 3.* Ablation Studies on Memory Routing

preprocessing de-contextualization, ensuring deep reasoning capabilities and robustness.

**Results on HotpotQA Benchmark.** As illustrated in Table 2, E-mem demonstrates exceptional stability in ultra-long context scenarios, consistently maintaining the highest F1 scores against all baselines. A particularly notable observation is the performance of RAG compared to other complex memory-based baselines, representing a clear divergence from the trends observed in the LoCoMo results. We attribute this discrepancy to the distinct information density inherent in the benchmarks: while LoCoMo features high-similarity, dense dialogues laden with adversarial noise that significantly impairs vector retrieval, HotpotQA comprises distinct, low-interference evidence passages that are structurally favorable for semantic matching. Nevertheless, E-mem still outperforms RAG by a substantial margin (+6.51% F1 in the 1600-doc setting). This confirms that while vector retrieval suffices for locating distinct factoids, the proposed episodic context reconstruction is indispensable for preserving the sequential dependencies required for complex multi-hop reasoning over extensive horizons.

### 4.3. Robustness Against Hallucination

To rigorously assess resilience against context noise, we conducted experiments on the LoCoMo adversarial subset by cross-evaluating varying Assistant scales and master backbones. Results (Table 3) demonstrate E-mem's robustness,

achieving a peak F1 of 95.74%. Crucially, our analysis reveals that optimal performance relies on the synergistic collaboration between both agents. While scaling the Assistant from 0.6B to 8B consistently enhances local episodic context reconstruction, the master agent's reasoning capability proves equally decisive. We observe significant performance variance across backbones, indicating that powerful global orchestration is indispensable for final reasoning and filtering hallucinations.

### 4.4. Ablation Study

**Backbone Sensitivity Analysis.** We evaluate model capacity by cross-evaluating master backbones and assistant scales (0.6B–14B) on LoCoMo conversation 1. As shown in Table 4, overall performance plateaus around the 4B mark (F1 ≈ 50.7%), indicating that small models suffice for basic episodic context reconstruction. However, detailed analysis reveals divergence across tasks. Scaling assistants to 8B and 14B significantly boosts Multi-hop reasoning (+9.45% over 4B). We attribute this to larger models' ability to discern implicit dependencies within memory contexts, bridging disparate evidence. Conversely, larger models suffer slight degradation in Single-hop tasks. We hypothesize this stems from over-reasoning, where models hallucinate complex correlations that distract from simple pattern matching. Smaller models rely on surface features, ensuring robustness for factual retrieval. Finally, master agent variations cause marginal fluctuations (<4%), confirming that E-mem's efficacy depends primarily on the Assistant's local reasoning.

**Impact of Memory Chunk Granularity**. We further investigate the sensitivity of E-mem to memory chunk size ($S_{chunk}$) under a constrained total memory budget (≈ 32K tokens) on the LoCoMo conversation 1. As shown in Table 5, performance exhibits an inverted U-shaped trajectory, peaking at $S_{chunk} = 8$K (F1=50.70%). When chunks are overly granular (e.g., 4K), the system retrieves an excessive number of fragmented contexts to fill the budget. This introduces significant semantic noise and irrelevant distractors, which overwhelms the reasoning process and leads

*Table 5.* Impact of memory chunks' size granularity on the performance of E-mem

| | Overall | | Single Hop | | Multi-Hop | | Temporal | | Open Domain | |
|---|---|---|---|---|---|---|---|---|---|---|
| | F1 | BLEU-1 | F1 | BLEU-1 | F1 | BLEU-1 | F1 | BLEU-1 | F1 | BLEU-1 |
| **4K** | 45.89 | 37.35 | 47.05 | 38.76 | 34.67 | 28.62 | 59.37 | 47.56 | 28.93 | 22.19 |
| **8K** | **50.70** | **40.93** | **49.46** | **41.21** | **42.66** | **35.38** | **66.91** | **51.45** | **31.08** | **23.15** |
| **12K** | 47.97 | 37.85 | 47.63 | 38.24 | 38.04 | 29.67 | 65.44 | 49.89 | 24.51 | 13.43 |
| **16K** | 47.08 | 37.19 | 44.81 | 36.25 | 37.18 | 32.83 | 66.24 | 50.79 | 29.11 | 14.30 |
| **32K** | 43.00 | 33.92 | 44.50 | 36.70 | 33.79 | 28.52 | 55.30 | 42.47 | 22.64 | 7.92 |

to retrieval failure. Conversely, overly large chunks (e.g., 32K) force the agent to process extended contiguous sequences, causing it to suffer from attention dilution or the "lost-in-the-middle" phenomenon, where critical details are overshadowed by the massive context. The 8K configuration strikes an optimal balance, preserving local context coherence while allowing for diverse evidence aggregation.

**Router Ablation Study.** We evaluate the scalability of the multi-pathway routing mechanism on the HotpotQA-1600 dataset. Empirical results (Figure 3a) indicate that activating a minimal subset of memory chunks (e.g., $k = 8$) is sufficient to surpass strong baselines. Crucially, we observe information saturation: increasing $k$ from 8 to 20 yields marginal gains despite the significantly larger retrieval volume. This phenomenon suggests that our router successfully filters noise, concentrating the vast majority of requisite semantic signals within the top-ranked candidates. It also serves as strong evidence for the system's scalability, since the routing mechanism can effectively isolate the most critical episodic dependencies within a compact window. By achieving near-optimal reasoning fidelity at low activation densities, E-mem avoids the computational overhead of processing extensive but redundant contexts, making it highly adaptable to large-scale deployment scenarios.

We further conduct an ablation study to evaluate the contribution of individual routing pathways. The results (Figure 3b) show that the full model achieves a peak F1 of 55.76. Removing Global Alignment ($\mathcal{P}_{\text{global}}$) precipitates the most significant drop to 45.30 ($\Delta - 10.46$), underscoring macroscopic narrative anchoring as the primary driver preventing context fragmentation. The Semantic Association ($\mathcal{P}_{\text{vec}}$) proves to be the second most critical component (47.90), demonstrating that high-dimensional vector alignment is essential for capturing implicit latent intents. Finally, excluding Symbolic Triggers ($\mathcal{P}_{\text{kw}}$) yields a moderate decline to 52.42. While less dominant, this pathway remains necessary for ensuring precise lexical grounding of specific entities. Overall, the consistent performance gaps confirm that these orthogonal signals are complementary, validating the efficacy of our Multi-Source Activation Union strategy.

**Multi-Agent Architecture**

To investigate whether a simpler pipeline—where retrieved

text chunks are directly fed into a single LLM—is sufficient, we conducted an ablation study on the LoCoMo benchmark by removing the Assistant agents. In this setting, the Master LLM directly processes the concatenated raw episodic contexts.

As shown in Table 6, the drastic performance collapse in the simpler pipeline confirms that our multi-agent design is fundamentally essential rather than an optional add-on. We attribute this to the following factors:

- **Cognitive Decoupling:** Forcing a single LLM to process concatenated raw text chunks requires it to simultaneously execute low-level detail extraction and high-level logical synthesis, leading to severe "attention dilution." Our MAS strictly decouples these tasks: Assistant agents specialize in local, noise-resistant extraction, while the Master agent focuses purely on global aggregation and planning.

- **Overcoming "Lost-in-the-Middle":** Concatenating multiple raw chunks often exceeds the effective reasoning window of standard models. Assistant agents process their assigned episodic contexts in complete isolation, performing **parallel local distillation** to shield the Master agent from raw distractor noise.

- **Cost Efficiency:** By delegating the heavy, complex reasoning of raw texts to multiple cost-effective Small Language Models (SLMs) and reserving the expensive Master LLM strictly for final aggregation, E-mem avoids context window explosion. This heterogeneous compute strategy significantly reduces token consumption and overall deployment costs.

### 4.5. Cost Effectiveness Analysis

To evaluate the deployment costs, we report the average number of tokens required per query on the LoCoMo. We categorize the computational overhead into Large Model Tokens ($T_L$, e.g., GPT-4o) and Small Model Tokens ($T_S$, e.g., Qwen3-4B). To quantify the overall cost, we introduce a Normalized Cost metric. Given the significant discrepancy in pricing and computational complexity between the two models, we adopt a highly conservative cost ratio of 1:10

*Table 6.* Ablation study on the LoCoMo benchmark comparing the proposed multi-agent system (MAS) with a simpler "direct-read" pipeline.

| Architecture | Overall F1 | Single-Hop | Multi-Hop | Temporal | Open Domain |
|---|---|---|---|---|---|
| Simpler Pipeline (Direct Read) | 38.27 | 48.83 | 30.62 | 31.87 | 18.55 |
| **E-mem (MAS)** | **54.17** | **59.23** | **42.64** | **59.82** | **24.89** |

*Table 7.* Cost-performance analysis on the LoCoMo dataset. We report the average token consumption per query alongside the accuracy metrics (F1 and BLEU-1).

| | F1 | BLEU-1 | $T_S$ | $T_L$ | Total Cost |
|---|---|---|---|---|---|
| **Long-Context** | 37.31 | 29.57 | none | 16910 | 169100 |
| **RAG** | 44.73 | 39.40 | none | 643 | 6430 |
| **A-mem** | 39.65 | 32.31 | none | 2520 | 25200 |
| **Mem0** | 45.1 | 34.92 | none | 973 | 9730 |
| **MEMORYOS** | 42.84 | 35.54 | none | 3874 | 38740 |
| **LIGHTMEM** | 38.44 | 34.37 | none | 612 | 6120 |
| **GAM** | 45.31 | 37.78 | none | 1254 | 12540 |
| **E-mem** | **54.17** | **44.34** | 2271 | **135** | **3621** |

$(c_S : c_L)$, assuming one large model token is equivalent to ten small model tokens in terms of the cost.

As shown in Table 7, full-context methods like Long-Context incur a prohibitive normalized cost (169k units), driven entirely by expensive large model processing. While retrieval baselines like RAG reduce this, they still depend on the large model for reasoning. In contrast, E-mem strategically offloads episodic context processing. Even under our conservative 1:10 assumption, E-mem reduces the normalized cost to approx. 3.6k units—a $43\times$ reduction compared to Long-Context—while achieving superior F1 performance. This demonstrates that E-mem not only delivers outstanding performance on accuracy but also economically viable for large-scale applications.

## 5. Conclusion

In this paper, we presented E-mem, a framework that shifts from destructive de-contextualization to episodic context reconstruction, which effectively preserves the contextual integrity essential for deep reasoning. Technically, we implement this by encapsulating episodic contexts managed by a heterogeneous hierarchical master-assistant architecture, which enables the precise reactivation of archived memory units within the active reasoning contexts. Extensive evaluations on both LoCoMo and HotpotQA benchmarks validate that our approach establishes a comprehensive performance lead, significantly outperforming existing preprocessing paradigms—particularly in complex multi-hop tasks—while maintaining low token cost via heterogeneous collaboration. We believe that E-mem serves as a vital complement to existing memory paradigms, offering a robust solution for high-precision, complex System 2 reasoning.

## Acknowledgements

This work was supported in part by National Key R&D Program of China No. 2024YFB2705300, NSFC Grant 62232011, 62402315, the Shanghai Science and Technology Innovation Action Plan Grant 24BC3201200, and the Shanghai Municipal Special Program for Basic Research on General AI Foundation Models Grant 2025SHZDZX026D09.

## Impact Statement

This paper introduces E-mem, a framework designed to enhance the long-context reasoning capabilities of LLM agents via episodic context reconstruction. Our work primarily aims to advance the reliability and precision of autonomous systems in complex domains such as legal forensics, scientific discovery, and medical diagnosis.

However, we acknowledge potential societal implications associated with memory-augmented agents. **First, the persistence of episodic memory raises privacy concerns regarding the storage and retrieval of sensitive user data.** While our hierarchical architecture separates low-level storage from high-level planning, rigorous data governance and access control mechanisms are essential for real-world deployment. **Second, as agents gain deeper reasoning capabilities (System 2), ensuring their alignment with human values and preventing the hallucination of false memories remains critical to mitigating safety risks.** We believe this work encourages further research into robust, accountable, and transparent memory systems for AI agents.

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

## A. Ablation Study: Lifelong Memory Accumulation

To directly evaluate lifelong memory accumulation, we extended the LoCoMo benchmark to a continuous setting spanning five consecutive conversations without memory resetting.

*Table 8.* Performance of E-mem on the LoCoMo benchmark with and without memory resetting across five continuous conversations.

| Setting | Overall F1 | Single-Hop | Multi-Hop | Temporal | Open Domain |
|---|---|---|---|---|---|
| Memory Reset (Original) | 54.17 | 59.23 | 42.64 | 59.82 | 24.89 |
| No Reset (Continuous) | 51.75 | 55.81 | 41.55 | 58.75 | 22.56 |

As shown in Table 8, E-mem incurs only a marginal degradation in the overall F1 score when exposed to cross-session noise. For comparison, we evaluated standard RAG and Mem0 under the identical no-reset setting and observed severe performance drops (from 44.76 to 33.65 for RAG, and from 45.10 to 37.03 for Mem0). These results strongly validate the robustness of E-mem in lifelong learning scenarios.

## B. Limitation & Trade-off

As shown in Table 9, E-mem achieves the most favorable accuracy–cost trade-off among all compared methods. Specifically, E-mem obtains the highest F1 score of 54.17, outperforming the strongest baseline GAM by 8.86 points, while also reducing the total cost from 12,540 to 3,621. This indicates that the proposed episodic context reconstruction does not simply improve performance by consuming more computation; instead, it reallocates computation from expensive global reasoning to lightweight local memory reconstruction, leading to better reasoning fidelity under a lower normalized cost. The main trade-off lies in latency. E-mem takes 11.25 seconds per query, which is higher than fast retrieval-oriented methods such as RAG, LightMem, and Mem0, but still lower than GAM. This latency comes from activating multiple episodic memory units and allowing assistant agents to reconstruct local evidence before master aggregation. Importantly, this is a deliberate design choice rather than an inefficiency: the additional inference-time coordination enables E-mem to preserve temporal dependencies and multi-hop evidence chains that are often weakened by purely retrieval-based memory systems.

*Table 9.* Cost–latency trade-off comparison on long-term memory evaluation. E-mem achieves the best F1 score and the lowest total cost, while incurring higher latency than lightweight retrieval-based methods.

| Method | F1 | Total Cost | Latency (s) |
|---|---|---|---|
| RAG | 44.73 | 4590 | 2.60 |
| LightMem | 38.44 | 6120 | **2.30** |
| Mem0 | 45.10 | 9730 | 2.17 |
| GAM | 45.31 | 12540 | 14.71 |
| E-mem | **54.17** | **3621** | 11.25 |

Therefore, E-mem should not be positioned as a universal replacement for conventional memory methods. Fast retrieval-based approaches remain suitable for simple factoid queries or real-time interaction scenarios. Instead, E-mem serves as a complementary memory mechanism for high-precision, complex long-term reasoning, especially when temporal consistency, context fidelity, and answer reliability are more important than minimal response latency.

## C. Validation of System 2 Reasoning Capabilities

We emphasize that disentangling complex state transitions amidst the dense noise of the LoCoMo benchmark inherently demands high-precision, System 2 reasoning. Our proposed E-mem framework is specifically engineered to address this challenge.

To further substantiate our claim regarding System 2 reasoning capabilities, we evaluated E-mem on the AMA benchmark[1]. This dataset features highly dynamic environments (e.g., gaming and embodied AI) that strictly necessitate explicit multi-step state tracking and deliberative planning.

*Table 10.* Performance comparison on the AMA benchmark across diverse dynamic environments. The results demonstrate E-mem's superiority in handling explicit state transitions.

| Method | Avg | SQL | Software | Game | Web | Embodied AI | Open World |
|---|---|---|---|---|---|---|---|
| Longcontext | 51.35 | 50.49 | 50.86 | 55.11 | 53.08 | 48.56 | 50.57 |
| A-mem | 32.24 | 33.33 | 29.17 | 38.44 | 42.16 | 21.11 | 28.89 |
| Mem0 | 21.06 | 12.97 | 22.32 | 27.42 | 40.25 | 11.94 | 16.67 |
| HippoRAG | 44.05 | 50.49 | 47.08 | 38.17 | 59.63 | 16.94 | 47.08 |
| **E-mem (Ours)** | **69.84** | **51.17** | **54.05** | **80.83** | **88.81** | **64.75** | **79.87** |

As demonstrated in Table 10, E-mem significantly outperforms baseline methods reliant on lossy preprocessing across all scenarios requiring multi-step state transitions. These results confirm that by preserving raw episodic contexts, E-mem successfully enables explicit deliberative planning.

## D. Strong Long-Context Baselines Comparison

We compared our multi-agent framework (Qwen3-4B + gpt-4o-mini) against strong single-model long-context approaches (gpt-4o and gpt-5.1).

| Method | Overall | Single-Hop | Multi-Hop | Temporal | Open Domain |
|---|---|---|---|---|---|
| GPT-4o (Long-Context) | 48.92 | 55.07 | 34.37 | 52.57 | 26.97 |
| GPT-5.1 (Long-Context) | 52.31 | 57.43 | **44.48** | 56.85 | **31.20** |
| **E-mem (Ours)** | **54.17** | **59.23** | 42.64 | **59.82** | 24.89 |

- **Analysis:** Relying solely on cost-effective small models, E-mem **outperforms GPT-4o** and achieves highly competitive results against the absolute SOTA **GPT-5.1** (surpassing it in Overall, Single-Hop, and Temporal tasks). This proves our framework empowers small models to match or beat expensive long-context giants.

## E. Graph baselines comparison

We also added graph baselines **AriGraph** (Anokhin et al., 2024) and **HippoRAG** (Gutiérrez et al., 2024) on Locomo benchmark.

| | Overall | Single-Hop | Multi-Hop | Temporal | Open Domain |
|---|---|---|---|---|---|
| RAG | 44.73 | 52.45 | 27.50 | 46.07 | 16.87 |
| AriGraph | 45.76 | 53.94 | 31.36 | 45.39 | 17.55 |
| HippoRAG | 46.91 | 52.77 | 31.89 | 50.65 | 21.11 |
| **Ours** | **54.17** | **59.23** | **42.64** | **59.82** | **24.89** |

By preserving raw episodic contexts, E-mem enables explicit deliberative planning, outperforming these graph-based preprocessing baselines.

---

[1] https://huggingface.co/datasets/AMA-bench

## F. Robustness against Document Ordering

We explicitly evaluated HotpotQA under random document shuffling.

| Setting | 400 | 800 | 1600 |
|---|---|---|---|
| Normal | 61.46 | 55.46 | 55.76 |
| Random | 63.05 | 54.60 | 56.17 |

The result supports that our gains stem from E-mem's routing and structure cross-segment, not arbitrary local co-location.

## G. Storage Overhead.

E-mem strictly decouples dormant memory from active computation. Although an active Assistant agent (e.g., Qwen-4B) requires approximately 10GB of VRAM, dormant agents are stored entirely on inexpensive disk storage (occupying only about 30KB per agent, which includes raw text, summaries, and embeddings). This design ensures near-zero scaling costs. We do **not** maintain persistent Key-Value (KV) caches. Instead, the system loads memory on-the-fly exclusively for agents explicitly activated by the routing mechanism. **While our system primarily operates in a highly memory-efficient raw-text mode, we also provide an optional KV-caching mode (incurring an additional overhead of approximately 2GB per active agent).** In VRAM-constrained environments, a single LLM instance can process these activated agents sequentially. This architecture strictly bounds the peak VRAM footprint regardless of the total memory archive size, achieving lifelong scalability by trading off latency.

## H. Algorithm Details

We provide the formal procedural description of the E-mem framework in Algorithm 1. This process encompasses three distinct phases: (1) *Memory Building*, where long memory is processed into persistent episodic contexts; (2) *Associative Activation*, where relevant units are identified via multi-pathway routing; and (3) *Episodic Context Reconstruction & Reasoning*, where assistant agents reconstruct memory contexts for local reasoning to generate evidence.

## I. Implementation Details of the Retriever and Routing Mechanism

**Retriever Setup.** To ensure high retrieval performance, all baseline methods in our experiments employ the powerful OpenAI `text-embedding-3-small` model. In contrast, the initial version of E-mem utilized the lightweight `all-MiniLM-L6-v2` model for routing. When E-mem is equipped with the same OpenAI embedding model, it achieves an F1 score of 55.46. This explicitly demonstrates that our performance gains are fundamentally driven by the architectural design rather than the capacity of the underlying retriever.

**Two-Stage Cascaded Routing Algorithm.** Due to space constraints in the main text, we detail the implementation of our two-stage cascaded routing algorithm here. The process operates as follows:

1. **Threshold Override:** If any individual score ($S_{vec}$, $S_{kw}$, or $S_{global}$) of a text chunk exceeds a predefined threshold, that chunk is immediately activated.

2. **Weighted Top-$k$ Fill:** For the remaining text chunks, we apply Min-Max normalization to their scores. These chunks are then ranked based on a weighted sum score, defined as $w_1 \tilde{S}_{global} + w_2 \tilde{S}_{vec} + w_3 \tilde{S}_{kw}$, to fill the remaining activation budget of $k$ chunks.

---

**Algorithm 1** E-mem: Episodic Context reconstruction Framework

---

    **Input:** Stream $X$, Query $q$, Window $L$, Stride $S$, Agents $\mathcal{A}_{master}, \mathcal{M}_{asst}$

    **Output:** Synergistic Response $R$

---

       *Phase 1: Memory Building & Encapsulation*

  1: $C \leftarrow$ SlidingWindow$(X, L, S)$

  2: **for** each chunk $c_i \in C$ **do**

  3:      $\mathcal{E}_i \leftarrow$ Encapsulate$(c_i)$            $\triangleright$ Create episodic context

  4:      $s_i \leftarrow$ Summarize$(\mathcal{E}_i)$            $\triangleright$ High-level summary

  5:      $v_i, k_i \leftarrow$ Index$(\mathcal{E}_i)$

  6:      Archive$(\mathcal{E}_i, s_i, v_i, k_i)$            $\triangleright$ Store as persistent unit

  7: **end for**

       *Phase 2: Associative Activation (Routing)*

  8: $\mathcal{A}^* \leftarrow \emptyset$

  9: $\mathcal{P}_{glob} \leftarrow$ GlobalAlign$(q, \{s_i\})$

10: $\mathcal{P}_{sem} \leftarrow$ SemSim$(q, \{v_i\})$

11: $\mathcal{P}_{kw} \leftarrow$ SymMatch$(q, \{k_i\})$

12: $\mathcal{A}^* \leftarrow \mathcal{P}_{glob} \cup \mathcal{P}_{sem} \cup \mathcal{P}_{kw}$            $\triangleright$ Union of activation paths

       *Phase 3: Episodic Context Reconstruction & Reasoning*

13: $E \leftarrow \emptyset$

14: **for** each agent $\mathcal{A}_k \in \mathcal{A}^*$ **do**

15:      ReconstructContext$(\mathcal{E}_k)$            $\triangleright$ Re-experience native context

16:      $e_k \leftarrow \mathcal{M}_{asst}.$LocalReason$(q \mid \mathcal{E}_k)$            $\triangleright$ Generate evidence

17:      $E \leftarrow E \cup \{e_k\}$

18: **end for**

       *Phase 4: Synergistic Response*

19: $R \leftarrow \mathcal{A}_{master}.$Aggregate$(q, E)$

20: **return** $R$

---

## J. Discussion and Future Work

### J.1. The Latency-Fidelity Trade-off

While E-mem excels in reasoning fidelity, we acknowledge the inherent trade-off in inference latency required to reconstruct the full episodic context. We frame this as a justifiable Latency-Fidelity trade-off. Current memory paradigms optimize for millisecond retrieval at the cost of context integrity, suiting simple factoid queries but failing at complex reasoning. In contrast, E-mem prioritizes the integrity of episodic memory context, making it ideal for high-precision, knowledge-intensive scenarios where the cost of logical errors far outweighs the latency overhead (Table **??**). Examples include:

- **Legal & Financial Forensics:** Analyzing long-horizon evidence where missed dependencies cause critical failures.

- **Medical Diagnosis:** Synthesizing patient history from disparate timelines with paramount precision.

- **Scientific Review:** Connecting latent hypotheses across papers without hallucination.

In these domains, higher latency is a tolerable cost for rigorous, hallucination-free deduction.

### J.2. Future Research: An Adaptive Hybrid Framework

Crucially, E-mem's episodic context reconstruction is orthogonal to traditional retrieval. Vector retrieval excels at broad semantic relevance, while episodic reconstruction enables deep contextual reasoning.

Our future work aims to synthesize these into a unified Adaptive Dual-Mode Framework that dynamically selects the mechanism based on task complexity:

1. **Fast Mode (database-centric):** Uses standard RAG for simple factoids or casual conversation, ensuring real-time responsiveness.

2. **Deep Research Mode (E-mem):** Activates E-mem for complex planning or multi-hop reasoning, performing deep episodic memory reconstruction for logical rigor.

This hybrid architecture will allow agents to seamlessly alternate between "System 1" (fast retrieval) and "System 2" (slow, deep reasoning), balancing user experience with high-fidelity memory demands.

## K. Case Study: Cross-Session Reasoning

To demonstrate the efficacy of E-mem's episodic context reconstruction mechanism compared to traditional memory baselines, we present a qualitative analysis using a sample from the LoCoMo dataset (Conversation 1). We selected a multi-hop reasoning question that requires linking temporal information from one session with entity attribute information from a subsequent session, the case study is shown on Figure 4.

**Case Study: Multi-Hop Reasoning on LoCoMo**

**QA**  "Where did Caroline move from 4 years ago?"

**Answer**  Sweden.

**Difficulty**  The temporal anchor ("4 years ago") and the action ("moved") appear in **Session 3**, but the specific location ("Sweden") is only revealed in **Session 4** in the context of a different topic (a necklace).

---

**1. Traditional Memory (Baseline)**

*Mechanism: Standard top-$k$ vector retrieval based on semantic similarity.*

- **Step 1: Retrieval (De-contextualization Failure)**

  - **Hit (High Similarity):** Retrieves `D3:13`: *"I've known these friends for 4 years, since I moved from my home country."* (Matches "move" and "4 years").
  - **Miss (Low Similarity):** Fails to retrieve `D4:3`: *"This necklace is super special... a gift from my grandma in my home country, Sweden."*
  - *Analysis:* The embedding for `D4:3` is dominated by semantics related to "necklace" and "gift". The vector distance to the query "moving" is too large, causing it to fall outside the top-$k$ window.

- **Step 2: Generation (Information Gap)** The LLM receives context `D3:13` but lacks the specific entity link found in `D4:3`.

- **Outcome: Failure.** *"Caroline moved from her home country 4 years ago, but the specific country is not mentioned."*

---

**2. E-mem (Our Approach)**

*Mechanism: Hierarchical Master-Assistant with State Restoration.*

- ✓ **Step 1: Multi-Pathway Activation** master agent analyzes query. Uses **Global Alignment** ($\mathcal{P}_{global}$) to identify narrative timeframe ("moving") and activates **Session 3**. Simultaneously, uses **Symbolic Trigger** ($\mathcal{P}_{kw}$) to scan for location entities like "country", activating **Session 4**.

- ✓ **Step 2: parallel State Restoration & Reasoning**

  - **assistant agent (Session 3):** Identifies temporal anchor: *"I moved from my **home country** 4 years ago"* (`D3:13`).
  - **assistant agent (Session 4):** Through contextual reasoning, identifies that *"grandma in my **home country, Sweden**"* (`D4:3`) provides the specific location detail required by the query.

- ✓ **Step 3: Synergistic Response** master agent aggregates evidence from multi assistants: (1) Moved 4 years ago from home country (Session 3) + (2) Home country is identified as Sweden (Session 4).

- ✓ **Outcome: Success.** *"Caroline moved from **Sweden** 4 years ago."*

*Figure 4.* E-mem's state restoration enables successful answering, whereas traditional database-centric vector retrieval methods fail.

# L. Detailed Prompt

## System Prompt for Assistant Agent

```
# ROLE: Memory Retrieval & Analysis Agent

You are an advanced **Memory Retrieval & Analysis Agent**. You will be provided
    with a large amount of memory segments, each separated by a newline. Your goal
    is to process User Inputs based on two distinct modes: **Retrieval Mode** (for
    questions) and **Summary Mode** (for summarization requests).

### 1. MEMORY CONTEXT
All the memory context are provided above. Please read them carefully before
    answering!

---

### 2. CORE INSTRUCTION & MODES

Analyze the User's input carefully to determine the intent.

#### MODE A: RETRIEVAL & ANSWERING (Triggered by specific questions)
If the user asks a question that requires specific details from the memory:

**Step 1: Extract Original Memories (CRITICAL)**
*   Scan the provided memory segments.
*   Identify segments that are **directly relevant** to answering the user's
    question.
*   **CONSTRAINT:** You must extract the text **VERBATIM (word-for-word)**. Do not
    summarize, paraphrase, merge, or fix typos in this step. The subsequent model
    relies on exact phrasing to find the source.
*   Select enough context to be complete, but strictly filter out irrelevant noise.

**Step 2: Formulate Preliminary Answer**
*   Based **ONLY** on the extracted memories from Step 1, formulate a direct answer
     to the user's question.
*   Explain your reasoning briefly.

**Step 3: Structured Output**
You must output the result in the following strict XML format:

<response_type>retrieval</response_type>
<relevant_memories>
    <memory_segment>Paste exact original text of segment 1 here</memory_segment>
    <memory_segment>Paste exact original text of segment 2 here</memory_segment>
    <!-- Add more segments if necessary -->
</relevant_memories>
<model_reasoning>
    Based on the segments above, the answer is: [Your direct answer and reasoning
    here].
</model_reasoning>

#### MODE B: SUMMARIZATION (Triggered by "summary", "recap", or "overview" commands)

If the user asks to summarize the memories:

**Step 1: Analyze Key Elements**
Identify the following strictly:
*   **Speakers/Entities:** Who is involved?
*   **Time Periods:** When did things happen?
*   **Main Events:** What are the core actions, discussions, or topics?
```

```
*   **Key Items/Objects:** What specific tools, documents, or objects were
    mentioned?

**Step 2: Generate Concise Summary**
Create a summary that is brief but captures all critical information identified
    above. Avoid fluff.

**Step 3: Structured Output**
You must output the result in the following strict XML format:

<response_type>summary</response_type>
<summary_content>
    <speakers>[List speakers/entities]</speakers>
    <time_period>[List relevant times/dates]</time_period>
    <key_items>[List important objects/items]</key_items>
    <main_events>
        [A concise narrative of what happened]
    </main_events>
</summary_content>

---

### 3. NEGATIVE CONSTRAINTS (MUST FOLLOW)
1.  **NO HALLUCINATION:** If the answer is not in the memory, strictly state in `<
    model_reasoning>` that information is missing. Do not invent facts.
2.  **NO MODIFICATION:** In `<relevant_memories>`, never change a single character
    of the source text.
3.  **NO OUTSIDE KNOWLEDGE:** Answer only based on the provided memory context. Do
    not use general world knowledge unless it helps interpret the text context.

---

### 4. ONE-SHOT EXAMPLE

**Memory Context:**
[2023-10-01 14:00] Alice: I put the red key in the top drawer.
[2023-10-01 14:05] Bob: Okay, I will take the blue folder to the meeting.
[2023-10-02 09:00] Alice: Did you see the red key? I moved it to the kitchen table
    later that night.

**User Input:** "Where is the red key?"

**Your Output:**
<response_type>retrieval</response_type>
<relevant_memories>
    <memory_segment>[2023-10-01 14:00] Alice: I put the red key in the top drawer.</
    memory_segment>
    <memory_segment>[2023-10-02 09:00] Alice: Did you see the red key? I moved it
    to the kitchen table later that night.</memory_segment>
</relevant_memories>
<model_reasoning>
    Although Alice initially put the key in the top drawer, she explicitly states
    later that she moved it to the kitchen table. Therefore, the red key is
    currently on the kitchen table.
</model_reasoning>
```

## Summary Instructions for Assistant Agent

```
You are a helpful assistant. You will be provided with multiple pieces of context
    information.
Read all of them carefully.

When user asks questions, you MUST provide specific INFORMATION based strictly on
    the provided context.
When user asks you to summarize all the information, you MUST summarize all the
    provided context.

###  INTELLIGENT THINKING AND UNDERSTANDING REQUIREMENT ###
When processing user questions and context segments, you MUST engage in intelligent
     thought and reasoning to genuinely understand the question and the original
    text's meaning and intent.

### NOTE when providing information ###
- Never Provide answers directly.
- The information you provide should be the ORIGINAL INFORMATION the user mentioned
     before, and you MUST ensure the information is provided based on your
    understanding of the question and the original text segment.
- **You MUST provide the original information relevant to the user's question and
    explain its meaning or significance within the current context.**
- Do not make assumptions beyond what is explicitly stated.
- Provide as many relevant information as possible.

### NOTE when summarizing information ###
- Summarize all the provided context accurately and concisely.
- Leave out any unimportant thing, but KEEP ALL the Useful details!
```

---

System Prompt for master agent

```
# ROLE: Memory Fact Aggregator & Logic Solver

You are a **Strict Information Aggregator**. You sit between the raw memory
    database and the final response agent.
Your task is to analyze retrieved fragments, resolve conflicts, and synthesize a **
    clean, factual, and strictly grounded answer**.

### INPUT DATA
You receive raw memory blocks (Archived & Recent).
*    **Input Text:** The raw memory content (Ground Truth).
*    **Local Inference:** Preliminary guesses (Use only as hints; you must VERIFY
    them against the raw text).

### CRITICAL LOGIC

**1. NOISE ELIMINATION**
*    If a block is irrelevant to the specific query, **IGNORE** it.
*    If no blocks contain the answer, explicitly state that in the <answer_core>.

**2. CONFLICT RESOLUTION (The "Latest State" Rule)**
*    If memories conflict (e.g., object location changes), the **LATER Timestamp**
    overrides the earlier one.
*    *Example:* [10:00] "Key in drawer" vs [10:05] "Key on table" -> Truth: "Key on
    table".

**3. MULTI-HOP SYNTHESIS (Connect the Dots)**
*    Sometimes you must combine information from different piece of information (
    some even cross blocks) to form a complete answer.
*    *Example:* Block A says "Bob picked up the apple." Block B says "Bob gave the
    item in his hand to Alice." -> **Synthesis:** "Bob gave the apple to Alice."

**4. STRICT TEXTUAL HANDLING (The "Original Phrasing" Rule)**
*    **Do NOT paraphrase specific terms.** If the memory says "Crimson Artifact", do
    not call it "Red Item". Use the exact unique nouns/verbs from the text.
*    **PRONOUN RESOLUTION:** You **MUST** replace pronouns like 'I', 'Me', 'We' with
    the actual Speaker's Name or 'the User' if the name is not specified (And
    expressed as 'I'/'Me') to make the sentence standalone and clear for the next
    agent.
*    **Implicit -> Explicit:** If an action implies a state (e.g., "put on the table
    "), describe the current state (e.g., "is on the table").

---

### OUTPUT FORMAT (Strict XML)

<aggregator_output>
    <!-- STEP 1: Copy EXACT text segments that prove your answer. -->
    <evidence_quotes>
        <quote timestamp="[Time]">[Verbatim text from memory]</quote>
        <quote timestamp="[Time]">[Verbatim text from memory]</quote>
    </evidence_quotes>

    <!-- STEP 2: Explain how you resolved conflicts or connected dots. -->
    <logic_trace>
        [E.g., "Block 2 (10:05) overrides Block 1 (09:00). Combined with object
    info from Block 3."]
    </logic_trace>

    <!-- STEP 3: The factual answer statement.
        - Must be a COMPLETE sentence (Subject + Verb + Object).
```

```
            - Must use processed names instead of "I".
            - NO conversational filler (e.g., "I think", "Based on the text").
            - Just the raw fact. -->
    <answer_core>
        [The synthesized factual statement.]
    </answer_core>
</aggregator_output>

---

### ONE-SHOT EXAMPLE

**Query:** "Who has the report and where is it?"

**Input:**
> *Block 1:* [2023-05-20 14:00] Alice: I printed the TPS report and put it on my
    desk.
> *Block 2:* [2023-05-20 14:10] Bob: I walked by Alice's desk and took the document
     she left there. I am heading to the archive room now.

**Your Output:**
<aggregator_output>
    <evidence_quotes>
        <quote timestamp="2023-05-20 14:00">printed the TPS report</quote>
        <quote timestamp="2023-05-20 14:10">took the document she left there</quote
    >
        <quote timestamp="2023-05-20 14:10">heading to the archive room now</quote>
    </evidence_quotes>
    <logic_trace>
        Block 1 identifies the document as "TPS report". Block 2 states Bob took it
     from the desk 10 mins later. Bob has it in the archive room.
    </logic_trace>
    <answer_core>
        Bob has the TPS report and he is in the archive room.
    </answer_core>
</aggregator_output>

---

### REAL INPUT
**Query:** {query}

**Raw Memory Results:**
{results}
```

