# OpenReview forum: "E-mem: Multi-Agent Based Episodic Context Reconstruction for LLM Agent Memory"
_ICML.cc/2026/Conference — ICML 2026 regular_

### Official Review · Reviewer_FSHQ · 2026-02-26

**Soundness:** 2
**Presentation:** 3
**Significance:** 3
**Originality:** 2
**Overall Recommendation:** 4
**Confidence:** 3

**Summary:**

The paper introduces E-mem, a multi-agent framework designed to manage long-context memory for LLM agents without relying on lossy compression techniques. Instead of converting context into embeddings or graphs, which often strips away important sequential details, E-mem preserves the raw episodic context and reconstructs it only when necessary. The architecture is hierarchical, featuring lightweight assistant agents that manage specific memory segments and a central master agent responsible for high-level planning and synthesis. To retrieve the right information, the system uses a multi-pathway routing mechanism that combines global summary alignment, semantic embedding similarity, and keyword matching. Experiments on LoCoMo and HotpotQA suggest that E-mem outperforms existing baselines in F1 scores while significantly reducing token costs compared to full-context approaches.

**Compliance With Llm Reviewing Policy:**

Affirmed.

**Key Questions For Authors:**

Can you provide an experiment where baselines are allowed comparable total inference compute (e.g., multiple retrieval-reasoning passes)? This would help clarify if the improvements are truly due to the episodic reconstruction design rather than just the additional processing steps.

**Limitations:**

yes

**Strengths And Weaknesses:**

Strengths

- The authors identify a genuine limitation in existing memory systems that preprocessing methods like graph construction or hierarchical indexing inherently lose fine-grained sequential dependencies. The proposed "episodic context reconstruction" approach offers a logical alternative that preserves the raw context necessary for complex reasoning.

- The multi-pathway routing mechanism is well-engineered. By combining three orthogonal activation pathways, the system effectively mitigates the failure modes associated with relying on a single retrieval method.

Weaknesses

- It is difficult to isolate the source of the performance gains. E-mem utilizes a multi-agent setup involving multiple inference passes (SLM assistants plus a Master LLM), whereas many baselines use a single model pass. It is unclear whether the superior performance stems from the novel episodic reconstruction paradigm or simply from the increased inference compute allocated to the task.

- While the paper acknowledges a "latency-fidelity trade-off," the actual runtime costs are worrying. The system is significantly slower than baselines like Lightmem, particularly during the reasoning phase. This raises valid concerns about whether the architecture is scalable or practical for real-time applications, and the current analysis of this limitation is somewhat qualitative.

---

> ### Author Rebuttal · Authors · 2026-03-30
>
> **Dear Reviewer FSHQ,**
>
> We sincerely thank you for recognizing E-mem as a "logical, well-engineered alternative." To address your insightful questions on compute fairness, we added new compute-matched and SOTA graph baselines, explicitly proving that our massive performance gains are strictly architecture-driven rather than merely compute-driven.
>
> **Response to W1 & Q1: Compute Fairness, Baselines, and Performance Isolation**
>
> **1. E-mem's performance vs. Baseline Complexity**
>
> We clarify that all methods employs identical model and retrieval settings in the experiment. E-mem's performance gain comes entirely from uncompressed, full-text chunk reasoning by small models, rather than multi-turn iterations. Conversely, many advanced baselines (e.g., A-mem, Mem0, GAM, MemoryOS) inherently rely on highly complex, multi-step LLM processing, iterative reflection, and multi-agent collaboration. Therefore, E-mem’s superiority stems from its architectural paradigm, not compute stacking.
>
> **2. New Compute-Matched & Structured Baselines**
>
> We evaluated representative baselines (RAG, Mem0, Lightmem) under **multi-turn settings (up to 3 passes)**, alongside new outstanding knowledge graph baselines (**AriGraph, HippoRAG**).
>
> | Method | Overall F1 | Single-Hop | Multi-Hop | Temporal | Open Domain |
> | :--- | :--: | :--: | :--: | :--: | :--: |
> | Lightmem (Multi-round) | 43.05 | 49.26 | 35.93| 45.25|20.82|
> | RAG (Multi-round) | 47.81 | 53.61 | 32.77 | 51.09 | 18.16 |
> | Mem0 (Multi-round) | 47.98 | 49.19 |39.84 | 51.26| 24.82 |
> | AriGraph | 45.76 | 51.72 | 31.36 | 45.39 | 17.55 |
> | HippoRAG | 47.81 | 52.77 | 31.88 | 50.65 | 21.11 |
> | **E-mem (Ours)** | **54.17** | **59.23** | **42.64** | **59.82** | **24.89** |
>
> **Analysis:** While multi-turn iterations and graph structures improve baseline scores, they still fall significantly short of E-mem. The root cause is their underlying lossy compression paradigm: preprocessing leads to permanent information loss, making it difficult for multi-turn retrieval to recover missing links. Conversely, retaining too much information during preprocessing introduces severe noise. E-mem bypasses this dilemma; its "local reasoning + global aggregation" paradigm evaluates raw contexts holistically, yielding optimal results.
>
> **3. Token Efficiency as Compute Optimization**
> In the LLM era, inference compute is strictly dictated by token consumption. E-mem is fundamentally designed to optimize this computational resource allocation. By delegating the heavy reading of raw texts to lightweight SLMs and reserving the expensive Master LLM strictly for evidence aggregation, E-mem achieves SOTA performance while significantly reducing computational costs.
>
> **Response to W2: Latency, Scalability, and Practicality**
>
> We deeply appreciate the reviewer's practical perspective.
>
> **Practicality and Real-Time Applications:** We clarify that E-mem is purposefully designed for **System 2 deliberative reasoning**, not System 1 real-time reflexes. For complex, multi-hop planning tasks where accuracy is paramount (System 2), a 10-15% absolute F1 gain vastly outweighs the investment of a few seconds in compute time.
>
> As discussed in Appendix B, in a practical deployment, E-mem is not meant to replace fast retrieval but to complement it via an **Adaptive Dual-Mode Framework**. Systems can route simple, real-time queries to fast baselines (like Lightmem or RAG) and dynamically activate E-mem's multi-agent framework exclusively for complex tasks requiring deep logical deduction. We will incorporate this quantitative breakdown and deployment paradigm into the main text to clarify its practicality.

---

> > ### Author Rebuttal · Reviewer_FSHQ · 2026-04-06
> >
> > I am satisfied with the rebuttal and maintain my original score.

---

> > > ### Author Response · Authors · 2026-04-06
> > >
> > > Dear Reviewer **FSHQ**,
> > >
> > > We sincerely thank you for your insightful and high-quality feedback, and we are honored that our rebuttal has successfully resolved all of your concerns.
> > >
> > > We will carefully incorporate all the valuable suggestions and discussion during the review process into the final version of our paper.
> > >
> > > Thank you once again for the time and effort you dedicated to reviewing our work, which has been truly crucial in refining and improving our manuscript.

---

### Official Review · Reviewer_ndW8 · 2026-03-04

**Soundness:** 3
**Presentation:** 2
**Significance:** 3
**Originality:** 2
**Overall Recommendation:** 4
**Confidence:** 3

**Summary:**

This paper studies long-term memory for LLM agents and argues that existing memory systems usually rely on preprocessing (e.g., embeddings or structured compression), which may break sequential dependencies and causal relationships in the original context.

To address this issue, the paper proposes E-mem, a framework that shifts from memory preprocessing to episodic context reconstruction.

Instead of compressing memory, the system preserves raw context segments and reconstructs relevant episodes during inference, where assistant agents perform local reasoning and a master agent aggregates the evidence.

**Compliance With Llm Reviewing Policy:**

Affirmed.

**Final Justification:**

The authors have addressed most of my concerns in the rebuttal and have added key experiments (such as cost-performance analysis and strong baseline comparisons), making the results more complete and convincing. As a result, my overall evaluation has improved.

**Key Questions For Authors:**

Is the main benefit of the proposed method primarily due to performing reasoning inside each retrieved memory chunk? Would a simpler pipeline that retrieves chunks and directly applies reasoning achieve similar performance?

What specific role do assistant agents play that cannot be replaced by a simpler architecture? Is the multi-agent design essential for the observed improvements?

What is the actual latency and computational overhead during inference when episodic reconstruction is applied?

**Limitations:**

yes

**Strengths And Weaknesses:**

## Strengths
- The paper studies an important problem: long-term memory and reasoning for LLM agents. The discussion on how memory preprocessing may damage contextual integrity is meaningful and well motivated.

- The episodic reconstruction idea is intuitive. Preserving full contexts and reconstructing them during reasoning is conceptually well aligned with multi-hop and temporal reasoning tasks.

- The proposed system achieves strong empirical improvements, reaching over 54% F1 on LoCoMo and improving over the previous state-of-the-art by about 7.75%.

## Weaknesses
- The overall architecture is quite complex, involving multiple agents, routing mechanisms, and episodic reconstruction modules, while the underlying idea can be summarized as retrieving memory chunks, performing local reasoning, and then aggregating the results.

- The necessity of the multi-agent design is not fully justified. It remains unclear whether a simpler retrieval-plus-reasoning pipeline could achieve similar performance.

- Episodic reconstruction requires reasoning over multiple memory segments, which may increase inference-time computational cost and latency.

---

> ### Author Rebuttal · Authors · 2026-03-30
>
> **Dear Reviewer ndW8,**
>
> We sincerely thank you for the insightful questions regarding the necessity of our multi-agent design and its computational overhead. We address them comprehensively below.
>
> **1. The Fundamental Necessity of the Multi-Agent System (MAS) (W1, W2, Q1, Q2)**
>
> **Response to Q1&W1.** E-mem pipeline: Retrieval, parallel local reasoning by SLMs, and global LLM aggregation.
>
> A simpler 'retrieve-then-reason' pipeline **fails to achieve comparable performance** because feeding massive concatenated chunks to a single LLM inevitably triggers the 'Lost-in-the-Middle' phenomenon due to severe context overload.
>
> **Response to Q2&W2.** To directly answer whether a simpler pipeline—retrieving chunks and directly applying reasoning via a single LLM—would suffice, we conducted an ablation study on LoCoMo by removing the Assistant agents:
>
> | Architecture | Overall F1 | Single-Hop | Multi-Hop | Temporal | Open Domain |
> | :--- | :--: | :--: | :--: | :--: | :--: |
> | Simpler Pipeline (Direct Read) | 39.71 | 48.83 | 30.62 | 31.87 | 18.55 |
> | **E-mem (MAS)** | **54.17** | **59.23** | **42.64** | **59.82** | **24.89** |
>
> The drastic performance collapse confirms that our multi-agent design is essential, not an optional add-on.
>
> Analysis：
>
> **Cognitive Decoupling:** A single LLM forced to read concatenated raw chunks must simultaneously perform low-level extraction and high-level logic synthesis, leading to severe "Attention Dilution." Our MAS strictly decouples these tasks: Assistants specialize in local, noise-resistant extraction, while the Master focuses purely on global aggregation and planning.
>
> **Overcoming "Lost-in-the-Middle":** Concatenating multiple raw chunks exceeds the effective reasoning window of standard models. Assistant agents process their assigned episodic contexts in complete isolation, performing **parallel local distillation** to shield the Master agent from raw distractor noise.
>
> **Cost Efficiency:** By delegating the heavy, complex reasoning of raw texts to multiple cost-effective small models, and reserving the expensive Master LLM strictly for final aggregation, E-mem avoids exploding context windows. This heterogeneous compute strategy significantly reduces token consumption and lowers overall deployment costs.
>
> **2. Inference Latency Breakdown (Response to W3, Q3)**
> While episodic reconstruction increases absolute latency compared to standard RAG, we deliberately designed this as a **Latency-Fidelity Trade-off** for rigorous System 2 reasoning. We break down the average 10.97s latency (e.g., HotpotQA-400, Appendix A, Table 7):
>
> **1) Routing (about 0.8s):** Efficient multi-pathway filtering.
>
> **2) Parallel Episodic Reconstruction (about 8.2s):** Assistants perform deep local
> reasoning. Executed in **parallel**, avoiding linear scaling bottlenecks.
>
> **3) Master Aggregation (about 2.0s):** Because Assistants pre-distill the noise, the Master’s context window is drastically reduced, enabling exceptionally fast synthesis.
>
> Trading 8.2s of parallel compute for a +14.46% F1 boost is highly pragmatic, and E-mem remains faster than the SOTA baseline GAM (12.43s). Furthermore, this architecture significantly reduces token cost compared to other methods (Table 6).
>
> **3. Orthogonality and the Adaptive Dual-Mode Ecosystem (Appendix B)**
> Importantly, we do not view E-mem as a blanket replacement for traditional methods, but rather as an **orthogonal** and highly optimized complement.
>
> * Standard RAG is akin to **System 1**: It excels at fast, pattern-matching semantic retrieval for simple factoids, ensuring real-time responsiveness.
>
> * E-mem represents **System 2**: It is designed specifically for complex planning and multi-hop reasoning, where traditional RAG fails (as evidenced by the 39.71 F1 score of the simple pipeline).
>
> As discussed in Appendix B, in real-world deployment, these two paradigms form an **Adaptive Dual-Mode Framework**. Systems can route simple queries to traditional methods for low latency, and activate our E-mem framework only when rigorous, hallucination-free logical deduction is required. Therefore, E-mem's computational overhead is a necessary, compartmentalized feature of a broader, intelligent agent ecosystem.

---

> > ### Author Rebuttal · Reviewer_ndW8 · 2026-04-04
> >
> > Thank you for the detailed response. The ablation results and the analysis of the multi-agent design are generally convincing, and they adequately address the concerns regarding necessity and latency.
> >
> > For the final version, I encourage the authors to further strengthen the presentation—within the review guidelines—to improve overall persuasiveness. In particular, more explicit quantification of the cost–performance trade-off and a clearer comparison with strong single-model / long-context baselines would be helpful.

---

> > > ### Author Response · Authors · 2026-04-04
> > >
> > > Dear Reviewer  **ndW8**
> > >
> > > We sincerely thank you for confirming that our rebuttal has addressed all concerns.
> > >
> > > Following your valuable suggestion, we have explicitly quantified the cost-performance trade-off and added comparisons against strong long-context baselines.
> > >
> > > **1. Cost-Performance Trade-off**
> > >
> > > We have summarized our existing data on the LoCoMo benchmark to report F1, Total Token Cost (see Section 4.5), and Latency..
> > >
> > > | Method | F1 Score | Total Cost | Latency (s) |
> > > | :--- | :---: | :---: | :---: |
> > > | RAG | 44.73 | 4590 | 2.60 |
> > > | Lightmem | 38.44 | 6120 | 2.30 |
> > > | mem0 | 45.10 | 9730 | **2.17**|
> > > | GAM | 45.31 | 12540 | 14.71 |
> > > | **E-mem (Ours)** | **54.17** | **3621** | 11.25 |
> > >
> > > **Analysis:** E-mem achieves the **highest F1 score** (+8.86 over GAM) with the **lowest total cost**. While episodic reconstruction adds latency compared to simple RAG, it remains faster than the strongest baseline, explicitly supporting the trade-off for high-fidelity reasoning.
> > >
> > > **2. Strong Long-Context Baselines**
> > > We compared our multi-agent framework (Qwen3-4B + gpt-4o-mini) against strong single-model long-context approaches (gpt-4o and gpt-5.1).
> > >
> > > | Method | Overall | Single-Hop | Multi-Hop | Temporal | Open Domain |
> > > | :--- | :---: | :---: | :---: | :---: | :---: |
> > > | GPT-4o (Long-Context) | 48.75 | 55.07 | 34.37 | 52.57 | 26.97 |
> > > | GPT-5.1 (Long-Context) | 53.31 | 57.43 | **44.48** | 56.85 | **31.20** |
> > > | **E-mem (Ours)** | **54.17** | **59.23** | 42.64 | **59.82** | 24.89 |
> > >
> > > * **Analysis:** Relying solely on cost-effective small models, E-mem **outperforms GPT-4o** and achieves highly competitive results against the absolute strong **GPT-5.1** (surpassing it in Overall, Single-Hop, and Temporal tasks). This indicates our framework empowers small models to match or beat expensive long-context giants.
> > >
> > > We will incorporate additional experiments and discussions (e.g., cost–performance quantification and stronger baselines) in the final version.
> > >
> > > We deeply appreciate the time you devoted to improving our work.

---

### Official Review · Reviewer_3pAy · 2026-03-13

**Soundness:** 3
**Presentation:** 2
**Significance:** 2
**Originality:** 3
**Overall Recommendation:** 4
**Confidence:** 4

**Summary:**

The paper presents E-mem, a memory architecture for LLM agents that aims to preserve raw episodic context and reconstruct relevant evidence at query time. Incoming text is segmented into overlapping windows, and assistant agents backed by a smaller language model generate summaries for each memory segment while retaining the full raw context. At query time, a hybrid routing mechanism combines summary alignment, embedding similarity, and keyword matching to select relevant memory segments. The selected assistant agents then reason locally over the retained raw contexts and produce candidate evidence, which a larger master model aggregates into a final answer. Experiments on LoCoMo and a streaming adaptation of HotpotQA report strong gains over the included baselines, along with lower normalized token cost.

**Compliance With Llm Reviewing Policy:**

Affirmed.

**Final Justification:**

I had initial concerns about the experimental setting of the paper, especially the streaming and memory-reset, retrieval setup at question-answering time, the scalability claims, and the absence of some structured-memory baselines. The rebuttal addressed several of these concerns meaningfully by adding a no-reset accumulation experiment, a document-order randomization test, additional structured-memory baselines, and larger-scale evaluations. At the same time, I still think the paper has important limitations, some of which I noted in my rebuttal acknowledgement. In particular, the no-reset experiment is still limited to five continuous conversations, so it only partially addresses the longer-term memory claim, and I also think the storage/overhead story and the “System 2 reasoning” framing should be clarified further in the final version. Overall, the rebuttal strengthened the empirical case enough to change my recommendation from 3 to 4, although important limitations still remain and should be stated more clearly in the revised paper.

**Key Questions For Authors:**

1. **Have you evaluated E-mem in a setting where memory persists and accumulates across tasks?** Both benchmarks appear to build and discard memory on a per-sample basis, so the central claims about lifelong learning and expanding operational horizons are not directly tested. Results in a setting where memory grows over time and queries span different ingestion phases would significantly strengthen the paper.

2. **How sensitive are the HotpotQA results to document ordering?** In the current setup, the fixed-seed shuffle may cause supporting documents to co-locate within a single 8K chunk. Have you tested whether performance remains stable under different randomized document orderings? An ablation along these lines would help address this possible confound.

3. **Can you clarify the exact routing implementation used in experiments?** The paper describes routing abstractly as a multi-pathway policy, but important implementation details such as pathway weighting, normalization, top-k selection, and any BM25-based override logic are not described clearly. Since these choices may materially affect performance, they seem necessary for reproducibility.

4. **What is the storage overhead per agent, and how does it scale with archive size?** The cost analysis focuses on token consumption, but E-mem also stores richer per-agent state than standard retrieval systems, including raw text, summaries, embeddings, retrieval indices, and potentially KV caches. For the lifelong-memory scenarios motivating the paper, how does this storage grow in practice?

5. **Why are graph-based or structured-memory baselines not included?** The paper argues that graph- or summary-based memory methods may lose important contextual structure, but the experiments do not include representative structured-memory baselines such as HippoRAG or GraphRAG. Comparing against such methods would help substantiate the claimed advantage of preserving richer episodic context, especially in the multihopQA scenario where I would expect them to show strong performance.

6. **How competitive is the retriever used in the retrieval-based baselines?** Since retriever quality is a major factor in the performance of methods like RAG, the paper should clarify which embedding model is used for retrieval and how sensitive results are to this choice. If the baselines use a weak retriever, part of the reported improvement may come from retrieval quality rather than from the proposed architecture itself.

**Limitations:**

Yes

**Strengths And Weaknesses:**

## Strengths
1. **Well-motivated core idea**: The paper is motivated by a real limitation of many memory methods: compressing context into summaries, embeddings, or graphs can discard sequential dependencies needed for temporal and multi-hop reasoning. Preserving richer episodic context and delegating local reasoning to assistant agents is a sensible architectural response.
2. **Targeted context delivery to the master model**: By having assistant agents extract localized evidence rather than forwarding entire raw chunks to the master model, the system is designed to reduce irrelevant context and focus downstream reasoning on more targeted information, which is plausibly helpful for issues like lost-in-the-middle effects and context rot noted by the authors. This is a practical and intuitively appealing design choice.
3. **Strong empirical results**: The paper reports strong results on LoCoMo across both master backbones, including particularly notable gains on multi-hop and temporal subsets. The HotpotQA results also show consistent improvements over the included baselines across all evaluated corpus scales.
4. **Promising token-cost efficiency**: The reported reduction in normalized token cost relative to baselines is practically relevant. The hierarchical design, in which the larger master model sees distilled evidence rather than full raw context, is an efficiency advantage.
5. **Thorough ablations**: The paper includes useful ablations on chunk granularity (Table 5), routing pathways (Figure 3b), backbone sensitivity (Table 4), and the number of activated chunks (Figure 3a). The hallucination analysis (Table 3) is also a strong addition.
6. **Helpful anonymous code release**: The anonymous code release was useful for understanding the implementation details of the method and improves the paper’s overall reproducibility. In particular, it helped clarify aspects of the system design that were only briefly described in the main text.

## Weaknesses
1. **Evaluation does not directly test persistent memory.** The paper frames E-mem as a memory system for “lifelong learning” and expanding operational horizons, but the evaluation does not directly test long-term accumulation in a persistent memory store. In the streaming HotpotQA setup, memory is constructed for each question and discarded afterward, making this primarily a long-context QA evaluation rather than a persistent-memory one. In LoCoMo, memory persists across questions within a conversation but is reset between conversations. As a result, the experiments do not show how the system behaves as memory accumulates over extended use.
2. **The “System 2 reasoning” framing is underdeveloped.** The term is used repeatedly as a central motivation, but the paper never clearly defines what System 2 reasoning means in this setting or explains how the benchmarks validate it. While the evaluated tasks involve memory retrieval, temporal reasoning, and some multi-hop inference, they do not clearly isolate broader deliberative capabilities such as planning or explicit multi-step decision making. This makes the System 2 framing feel more rhetorical than experimentally grounded.
3. **Important routing details are omitted from the paper.** The routing mechanism is described abstractly as a multi-pathway policy, but key implementation details appear to be missing from the paper and appendix. Based on the anonymous code release, the actual implementation involves pathway weighting, normalization, top-k filtering, and additional heuristics such as a BM25-based boost. These details may materially affect performance and should be described more explicitly for reproducibility.
4. **The “multi-agent” characterization may be overstated.** In the recommended shared-model setting, the assistant agents appear to share the same underlying smaller model and differ primarily in the local memory contexts they operate over. This makes the system look closer to a routed retrieve-extract-aggregate pipeline than to the richer multi-agent architecture suggested by the paper’s framing. The paper would benefit from a more precise characterization of what is gained specifically from the agent formulation.
5. **The HotpotQA setup may partially blur retrieval and multi-hop reasoning.** Given the chunk size used in the paper and the typical spacing between supporting documents in the constructed stream, it seems plausible that some gold evidence pairs may frequently co-occur within a single assistant’s context window. If so, part of the reported multi-hop gain could come from local co-location of evidence rather than from routing and aggregation across distinct memory segments. The paper does not analyze sensitivity to document ordering or evidence spacing, making this possible confound difficult to assess.
6. **Storage overhead is not analyzed.** The efficiency discussion focuses on token cost, but E-mem also stores substantially richer per-segment state than standard retrieval systems, including raw text, summaries, embeddings, retrieval indices, and in some operating modes KV caches. For a system framed as persistent or lifelong memory, storage overhead is an important practical consideration and is not discussed in the evaluation.
7. **Scalability is only evaluated at modest archive sizes.** The experiments scale to 1,600 documents, but the paper does not evaluate behavior at larger memory sizes where routing, storage, and latency may become more significant concerns. Since the routing stage appears to score all agents for each query, it would be useful to understand how performance and efficiency change at substantially larger scales especially as the number of required assistant agents starts to grow.

---

> ### Author Rebuttal · Authors · 2026-03-30
>
> **Dear Reviewer 3pAy,**
>
> We deeply appreciate your constructive feedback and high-quality questions on scalability, long-term memory, and baselines.
>
> **1. Memory Accumulation in lifelong memory (W1, Q1)**
>
> We extended the LoCoMo evaluation across 5 continuous conversations without memory resetting.
>
> || Overall | Single-Hop | Multi-Hop | Temporal | Open Domain |
> | :--- | :--: | :--: | :--: | :--: | :--: |
> | Memory Reset | 54.17 | 59.23 | 42.64 | 59.82 | 24.89 |
> | No Reset | 52.65 | 55.81 | 41.55 | 58.75 | 22.56 |
>
> E-mem yields a slight F1 drop against cross-session noise. **To compare, we explicitly ran RAG and Mem0 under the same setting, observing severe degradation (44.76 $\rightarrow$ 35.19 and 45.10 $\rightarrow$ 37.03)**, which supports our lifelong robustness.
>
> **2. Robustness against Document Ordering (W5, Q2)**
>
> We explicitly evaluated HotpotQA under random document shuffling.
>
> | Setting | 400| 800| 1600|
> | :--- | :--: | :--: | :--: |
> | Normal | 61.46 | 55.46 | 55.76 |
> | Random | 63.05 | 54.60 | 56.17 |
>
> The result supports that our gains stem from E-mem's routing and structure cross-segment , not arbitrary local co-location.
>
> **3. Structured Memory Baselines & Retriever Model (Q5, Q6)**
>
> **Retriever Setup:** Baselines use powerful OpenAI's text-embedding-3-small. E-mem initially uses all-MiniLM-L6-v2 for routing. **Equipping E-mem with the same OpenAI retriever yields 55.46 F1 (V.S. 54.17)**, supporting our gains are architecture-driven, not retriever-dependent.
>
> Per your advice, we added graph baselines **AriGraph** and **HippoRAG**.
>
> |  | Overall| Single-Hop | Multi-Hop | Temporal | Open Domain |
> | :--- | :--: | :--: | :--: | :--: | :--: |
> | RAG | 44.73 | 52.45 | 27.50 | 46.07 | 16.87 |
> | AriGraph | 45.76 | 51.72 | 31.36 | 45.39 | 17.55 |
> | HippoRAG| 47.81 | 52.77 | 31.88 | 50.65 | 21.11 |
> | **E-mem** | **54.17** | **59.23** | **42.64** | **59.82** | **24.89** |
>
> By preserving raw episodic contexts, E-mem enables explicit deliberative planning, outperforming these graph-based preprocessing baselines.
>
> **4. System Overhead (W4, W6, Q4)**
>
> **Storage Overhead (W6, Q4):** While an Assistant (e.g., Qwen-4B) needs 10GB VRAM, E-mem decouples dormant memory (30KB/agent on cheap disk in total including text, summary, embeddings...) from active compute, ensuring very low scaling costs. We eschew persistent KV caches, loading memory on-the-fly only for routed agents (defaulting to raw text, with an optional KV mode adding 2GB/agent). For VRAM-constrained setups, a single LLM instance can process these active agents sequentially. This bounds VRAM regardless of archive size, trading latency for scalability.
>
> **"Multi-Agent"(W4):** Instances with identical parameters but distinct contexts inherently constitute distinct agents. E-mem uses distinct agents to reason over separate memory segments. By having Assistants perform deep local reasoning on different episodes and a Master aggregate these insights, this division of labor embodies a true multi-agent collaboration paradigm.
>
> **5. System 2 (W2)**
>
>  **We clarify that processing complex state transitions amidst the dense noise of LoCoMo naturally requires high-precision, System 2 reasoning—a challenge E-mem is ideally suited to address.**
>
>  To further validate our System 2 claim, we evaluated E-mem on the AMA (https://huggingface.co/datasets/AMA-bench), which features highly dynamic environments (e.g. Games, Embodied AI) requiring explicit state transitions.
>
> || Avg |SQL | Software | Game | Web | Embodied AI | Open World|
> | :--- | :--: | :--: | :--: | :--: | :--: | :--: | :--: |
> | Longcontext | 51.35 | 50.49 | 50.86 | 55.11 | 53.08 | 48.56 | 50.57 |
> | A-mem | 32.24 | 33.33 | 29.17 | 38.44 | 42.16 | 21.11 | 28.89 |
> | Mem0 | 21.06 | 12.97 | 22.32 | 27.42 | 40.25 | 11.94 | 16.67 |
> | HippoRAG  | 44.05 | 50.49 | 47.08 | 38.17 | 59.63 | 16.94 | 47.08 |
> | **E-mem**| **69.84** | **51.17** | **54.05** | **80.83** | **88.81** | **64.75** | **79.87** |
>
> Result shows in scenarios requiring multi-step state transitions, E-mem outperforms lossy preprocessing baselines by preserving raw episodic contexts to enable explicit deliberative planning.
>
> **6. Routing Implementation Details (W3, Q3)**
> Omitted due to space limits, our routing algorithm operates as follows:
>
> 1) **Threshold Override:** Directly activates chunks if any single score ($S_{vec}, S_{kw}, S_{global}$) exceeds a predefined threshold.
>
> 2) **Weighted Top-k Fill:** Min-Max normalizes remaining chunks and ranks them via weighted sum ($W_1 S_{vec},+W_2 S_{kw},+W_3 S_{global}$) to fill the k budget.
>
> **7. Scalability to Ultra-Long Contexts (W7)**
>
> We extended the HotpotQA streaming evaluation up to 6,400 documents.
>
> | Metric | 400| 800 | 1600 | 3200 | 6400 |
> | :--- | :--: | :--: | :--: | :--: | :--: |
> | F1 | 61.46 | 55.46 | 55.76 | 52.75 | 54.38 |
>
> The result supports that the performance remains stable even at extreme context scales.

---

> > ### Author Rebuttal · Reviewer_3pAy · 2026-04-03
> >
> > I thank the authors for the detailed rebuttal and for performing meaningful new experiments. The additional evidence strengthens the paper, especially the no-reset memory experiment, the document-shuffling study and the added structured-memory baselines.
> >
> > These results address several of my main empirical concerns. I still have some reservations about the storage / systems-overhead and about the breadth of the “System 2 reasoning” framing, which I do not think is yet defined precisely enough. More broadly, I still think longer-term memory remains underexplored in this work: in a practical memory architecture, decisions about what to store, how to represent it, how to update it over time, and how to manage growth are all important design questions, while the current work primarily studies retrieval over an accumulated store. I view these as important limitations of the current paper, and I encourage the authors to address them explicitly in the final version.
> >
> > Even so, the additional experiments and clarifications in the rebuttal strengthen the empirical case, and I am therefore increasing my score from **3 to 4**.

---

> > > ### Author Response · Authors · 2026-04-04
> > >
> > > Dear Reviewer **3pAy**
> > >
> > > We sincerely thank you for the constructive feedback and for recognizing the value of our rebuttal experiments. All points raised during the review process will be discussed in greater depth in the final version. We are grateful for yours guidance throughout this process.

---

### Official Review · Reviewer_jwrb · 2026-03-13

**Soundness:** 3
**Presentation:** 3
**Significance:** 3
**Originality:** 3
**Overall Recommendation:** 5
**Confidence:** 4

**Summary:**

The paper proposes E-mem, a memory architecture for LLM agents that argues against aggressive memory preprocessing and instead preserves raw episodic segments that can later be selectively reactivated and reasoned over. The system uses a hierarchical master–assistant design: a master agent performs global planning and synthesis, while assistant agents each retain an uncompressed memory segment plus a summary used for routing. Query-time access is handled by a multi-pathway router combining summary-level alignment, embedding-based semantic association, and keyword matching; activated assistants then perform local reasoning over their preserved context and return temporally anchored evidence to the master for aggregation. Empirically, the paper reports improvements over several recent memory baselines on LoCoMo and on a long-context adaptation of HotpotQA, together with substantially reduced normalized token cost relative to full-context processing and standard retrieval baselines.

**Compliance With Llm Reviewing Policy:**

Affirmed.

**Final Justification:**

Authors have largely addressed my concerns.

**Key Questions For Authors:**

How does E-mem differ conceptually and operationally from AriGraph-style episodic memory?

How exactly is the HotpotQA long-context streaming benchmark constructed?

**Limitations:**

yes

**Strengths And Weaknesses:**

Strengths

The paper studies an important problem: how LLM agents should retain and recover long-horizon experience for temporally structured and multi-hop reasoning. The central idea of reserving raw episodic context and deferring compression until query time is intuitively sensible, and the reported gains on LoCoMo are substantial, especially on multi-hop and temporal subsets. The architecture is also reasonably coherent: memory construction, routing, assistant-side reconstruction, and master-side synthesis form a clear pipeline. The paper includes several useful analyses, including routing ablations, chunk-size sensitivity, and a cost comparison, which makes the empirical section broader than a minimal benchmark table.

Weaknesses

My main concern is that the paper overstates the novelty and does not position itself adequately with respect to closely related episodic-memory work. In particular, the core idea of organizing experience as episodic memory and recovering context for downstream reasoning appears closely related to AriGraph’s episodic memory formulation, yet AriGraph is neither referenced nor discussed, and no comparison is provided. This omission is important because the paper presents its framing as if the key conceptual move of episodic preservation rather than purely static preprocessing were largely novel. At minimum, the related-work section should acknowledge this line of work and explain the precise distinction between E-mem and prior episodic-memory architectures.

The HotpotQA streaming adaptation is insufficiently detailed to evaluate whether the setup is neutral.

Soundness

Soundness, the method seems plausible and the empirical results are meaningful. The HotpotQA streaming adaptation is insufficiently detailed to evaluate whether the setup is neutral.

Presentation

The paper is mostly readable and the high-level narrative is easy to follow, but the literature positioning needs revision. The omission of a clearly relevant prior work such as AriGraph is not a minor citation gap. The paper should more carefully separate what is new in E-mem from what is inherited from prior episodic or graph-based agent-memory designs.

Significance

I view the paper as moderately significant. Long-term memory for agents is an important topic, and the benchmark improvements suggest the system may be useful for memory-intensive QA and agent settings. But the impact is limited by the current lack of precise attribution and by the incomplete comparison to nearby prior work.

Originality

The paper appears novel primarily at the level of system packaging and engineering combination, not at the level of foundational concept. That distinction should be made explicit.

Overall, this is a promising and empirically strong paper, but the current version needs a more complete related-work treatment, especially around episodic-memory antecedents.

---

> ### Author Rebuttal · Authors · 2026-03-30
>
> **Dear Reviewer jwrb,**
>
> We sincerely thank the reviewer for the constructive feedback and the encouraging recognition that our method is "intuitively sound" with "substantial gains." Your insightful comments on literature positioning and benchmark construction have greatly helped us enhance the rigor and clarity of our paper. Below, we address your core concerns in detail.
>
> **1. Response to Q1 & W1: Conceptual and Operational Differences from AriGraph**
>
> We thank the reviewer for raising this important point regarding AriGraph.
>
> However, conceptually, AriGraph fundamentally remains a graph-based paradigm constrained by preprocessing and information compression.
>
> While both works share the motivation of incorporating episodic memory, E-mem and AriGraph **differ in their conceptual philosophies and operational architectures**:
>
> * **Conceptual Distinction: Triplet-Bottlenecked vs. Pure Preservation.** AriGraph elegantly integrates episodic vertices into a semantic graph. However, its access to raw episodic memories is fundamentally **bottlenecked by semantic preprocessing**. In AriGraph’s retrieval algorithm, episodic search strictly requires first successfully matching semantic triplets. If the LLM fails to extract a nuanced sequential dependency into a rigid triplet (subject, relation, object) during ingestion, that raw memory becomes virtually irretrievable. E-mem, conversely, embraces pure episodic preservation. By avoiding rigid graph extraction, our multi-pathway router directly accesses uncompressed contexts, ensuring no latent dependency is lost.
>
> * **Operational Distinction: Passive Search vs. Active Reconstruction.** Operationally, AriGraph navigates a pre-built graph via BFS-style traversal. E-mem treats memory access not as a passive retrieval, but as an *active generative reconstruction task*: activated Assistant SLMs directly read the raw contexts to infer and reconstruct temporally anchored evidence on-the-fly.
>
> * **Empirical Validation.** To ensure a fair comparison, we faithfully re-implemented AriGraph's core retrieval pipeline, adapting it to the LoCoMo conversational
> setting with the same backbone model (GPT-4o-mini) .
>
> | Method | Overall F1 | Single-Hop | Multi-Hop | Temporal | Open Domain |
> | :--- | :--: | :--: | :--: | :--: | :--: |
> | Standard RAG | 44.73 | 52.45 | 27.50 | 46.07 | 16.87 |
> | AriGraph | 45.76 | 51.72 | 31.36 | 45.39 | 17.55 |
> | **E-mem (Ours)** | **54.17** | **59.23** | **42.64** | **59.82** | **24.89** |
>
> As anticipated, the graph structure improves upon standard RAG (Multi-hop F1: 31.36 vs. 27.50). Crucially, E-mem establishes a significant margin over AriGraph (42.64 vs. 31.36). This explicitly confirms that E-mem's dynamic agentic reconstruction successfully avoids the "destructive de-contextualization" vulnerabilities inherent in graph-based triplet extraction, validating our unique architectural novelty.
>
> We will include this crucial baseline comparison and the in-depth discussion in the Camera-Ready version.
>
> **2. Response to Q2 & W2: Construction and Neutrality of the HotpotQA Streaming Benchmark**
>
> Due to page constraints, this specification was deferred.
>
> Now we clarify that **we did not construct this streaming setup using proprietary or potentially biased heuristics.**
>
> **Benchmark Source & Mechanism:** We adopted a rigorous, publicly available long-context adaptation of HotpotQA developed by researchers from Tsinghua University (https://huggingface.co/datasets/BytedTsinghua-SIA/hotpotqa/tree/main). This benchmark systematically injects varying volumes of irrelevant distractor documents into the original HotpotQA examples to create ultra-long contexts of specific scales (e.g., 400, 800, 1600 documents). The gold supporting facts are embedded within this massive sea of noise.
>
> Specifically, each question retains its gold paragraphs, with TF-IDF-retrieved Wikipedia distractors injected and shuffled to scale contexts to 400/800/1600 documents. All methods received identical inputs with no additional preprocessing, ensuring a fully neutral comparison.

---

> > ### Author Rebuttal · Reviewer_jwrb · 2026-04-03
> >
> > Authors have largely addressed my concerns.

---

> > > ### Author Response · Authors · 2026-04-04
> > >
> > > Dear Reviewer **jwrb**
> > >
> > > We sincerely thank you for the positive assessment and for the time spent evaluating our work. We are glad that the rebuttal has adequately addressed the concerns, and we will ensure the final version incorporates all the feedback received during the review process.

---

### Decision · Program_Chairs · 2026-04-30

**Decision:**

Accept (regular)

**Comment:**

E-mem preserves raw episodic context and defers compression to query time via a master–assistant multi-agent architecture, challenging the lossy preprocessing paradigm of prior memory systems. All four reviewers recognized the well-motivated framing, intuitive design, and strong gains on LoCoMo with reduced token cost. The rebuttal substantively addressed the main concerns — most notably compute fairness and the necessity of the multi-agent design, supported by compute-matched baselines and an Assistant-ablation showing a sharp drop without the assistants. Residual concerns around the strength of the "lifelong memory" claim and the operational definition of "System 2 reasoning" should be clarified in the camera-ready. All reviewers converged to accept; I recommend acceptance.